

# Modelling dynamic interactions between soil structure and the storage and turnover of soil organic matter

Katharina Hildegard Elisabeth Meurer[1]; Claire Chenu[2]; Elsa Coucheney[1]; Anke Marianne Herrmann[1]; Thomas Keller[1,3]; Thomas Kätterer[4]; David Nimblad Svensson[1]; Nicholas Jarvis[1]

[1] Swedish University of Agricultural Sciences, Department of Soil & Environment, SE-750 07 Uppsala, Sweden
[2] AgroParisTech, UMR Ecosys INRA-AgroParisTech, Université Paris-Saclay, F-78850 Thiverval-Grignon, France
[3] Agroscope, Department of Agroecology & Environment, CH-8046 Zürich, Switzerland
[4] Swedish University of Agricultural Sciences, Department of Ecology, SE-750 05 Uppsala, Sweden

*Correspondence to:* K. H. E. Meurer katharina.meurer@slu.se





**Abstract**

Models of soil organic carbon (SOC) storage and turnover can be useful tools to analyze the effects of soil and crop management practices and climate change on soil organic carbon stocks. The aggregated structure of soil is known to protect SOC from decomposition, and thus influence the potential for long-term sequestration. In turn, the turnover and storage of SOC affects soil aggregation, physical and hydraulic properties and the productive

capacity of soil. These interactions have not yet been explicitly considered in modelling approaches. In this study, we present and describe a new model of the dynamic feedbacks between SOM storage and soil physical properties (porosity, pore size distribution, bulk density and layer thickness). A sensitivity analysis was first performed to understand the behaviour of the model. The identifiability of model parameters was then investigated by calibrating the model against a synthetic data set. This analysis revealed that it would not be possible to unequivocally estimate

all of the model parameters from the kind of data usually available in field trials. Based on this information, the model was tested against measurements of bulk density and SOC concentration, as well as limited data on soil water retention and soil surface elevation, made during 63 years in a field trial located near Uppsala (Sweden) in three treatments with different OM inputs (bare fallow, animal and green manure). The model was able to accurately reproduce the changes in SOC, soil bulk density, surface elevation and soil water retention curves

observed in the field. Treatment-specific variations in SOC dynamics caused by differences in OM input quality could be simulated very well by modifying the value for the OM retention coefficient $\varepsilon$ (0.37 for animal manure and 0.14 for green manure). The model approach presented here may prove useful for management purposes, for example, in an analysis of carbon sequestration or soil degradation under land use and climate change.

**1 Introduction**

As a consequence of intensive cultivation, most agricultural soils have lost ca. 25–75 % of their antecedent store of SOC (Lal, 2013; Sanderman et al., 2017). Apart from contributing to the increase in atmospheric $CO_2$, this has also degraded the inherent physical quality and productivity of soil (e.g. Lal, 2007; Rickson et al., 2015; Henryson et al., 2018). This is because many important soil physical and hydraulic (e.g. water retention and hydraulic conductivity) properties are strongly influenced by soil organic matter (SOM). For example, SOM increases

porosity and reduces soil bulk density (e.g. Haynes and Naidu, 1998; Ruehlmann and Körschens, 2009; Jarvis et al., 2017). This is partly because the density of organic matter is less than that of soil minerals, but more importantly, it is a consequence of the aggregated soil structure induced by the microbial decomposition of fresh organic matter (Tisdall and Oades, 1982; Young and Crawford, 2004; Cosentino et al., 2006; Feeney et al., 2006; Bucka et al., 2019). Changes in the SOM content may also affect the pore size distribution, although the magnitude

of these effects across different ranges of pore diameter is still a matter of some controversy (e.g. Hudson, 1994; Rawls et al., 2003; Loveland and Webb, 2003; Minasny and McBratney 2018; Libohova et al., 2018).

The relationship between SOM and soil pore space properties can be characterized as a dynamic two-way interaction. This is because, in addition to the effects of SOM on soil pore size distribution and porosity, decomposition rates of SOM are reduced within microporous regions of soil that are poorly aerated and where the

carbon is physically much less accessible to microorganisms (e.g. Ekschmitt et al., 2008; Dungait et al., 2012; Lehmann and Kleber, 2015). Whereas sorption interactions with mineral surfaces are probably the dominant mechanisms protecting SOM from decomposition in coarse-textured soils, the additional physical protection afforded by microporous regions of the soil may lead to an enhanced long-term storage of SOM in structured fine-



textured soils (e.g. Hassink et al., 1993; Chevallier et al., 2004; Souza et al., 2017; Dignac et al., 2017). Thus, the
turnover of both particulate and soluble SOM has been shown to depend on its location in soil pore networks of
different diameter and connectivity and with contrasting microbial communities (e.g. Strong et al., 2004; Ruamps
et al, 2011; Nunan et al., 2017). Recent studies using novel X-ray imaging techniques have also provided additional
insights into how the soil pore space architecture regulates the physical protection of SOM in structured soil
(Kravchenko and Guber, 2017). For example, Kravchenko et al. (2015) showed that the decomposition rates of
intra-aggregate particulate SOM were 3 to 15 times faster in the presence of connected networks of aerated soil
pores > 13 μm in diameter than in the absence of such pores. Toosi et al. (2017) showed that plant residues
decomposed more slowly in soil microcosms dominated by pores 5-10 μm in diameter than in those containing a
significant proportion of pores > 30 μm in diameter. Quigley et al. (2018) showed that pores 40–90 μm in size
were associated with a fast influx of fresh carbon followed by its rapid decomposition, whereas soil pores < 40 μm
in diameter were associated with reduced rates of carbon decomposition. From the foregoing, it follows that the
turnover of SOM will be significantly affected by any physical or biological mixing process which transfers SOM
between different pore regions in soil. For example, soil tillage may promote decomposition by exposing SOM
that was previously effectively protected from microbial attack within microporous regions of the soil (e.g.
Balesdent et al., 2000; Chevallier et al., 2004). Physical protection of SOM is also affected by the mixing resulting
from the ingestion and casting of soil by earthworms (e.g. Martin, 1991; Görres et al. 2001; Angst et al., 2017).

Some widely-used models of SOM turnover and storage attempt to implicitly account for the effects of chemical
and physical protection by introducing a stable or inert pool (e.g. Falloon and Smith, 2000; Barré et al., 2010).
Other models have also been proposed that explicitly predict the effects of soil structure on SOM storage and
turnover by making use of the concept of soil micro- and macro-aggregates (e.g. Stamati et al., 2013; Segoli et al.,
2013). An alternative approach would be to define soil structure in terms of the soil pore space. The advantage of
this is that it allows a straightforward coupling to models of flow and transport processes in soil (e.g. Young et al.,
2001; Rabot et al., 2018). From a mathematical point of view, soil structure can be concisely described by the
volume and connectivity of solids and pore space and the surface area and curvature of their interface, all expressed
as a function of pore diameter (Vogel et al. 2010). We focus here on the total porosity and the pore size distribution,
since these properties underlie widely-used soil hydrological models based on Richards' equation. Incorporating
such a pore-space based approach to the interactions between SOM and soil structure into a soil-crop model would
enable explicit recognition of the feedback links that exist between SOM dynamics, soil hydrological processes
and plant growth (Henryson et al., 2018). Kuka et al. (2007) earlier proposed a pore-based model of SOM turnover
(CIPS), although they did not account for any feedbacks to soil physical properties and hydraulic functions. Here,
we propose and test a new model that describes the dynamic two-way interactions between SOM storage and
turnover, soil structure and soil physical properties. We first performed a sensitivity analysis of the proposed model
and also investigated parameter identifiability using a synthetic data set (e.g. Luo et al., 2017). This was done
because the data usually available from field experiments for testing models of SOM storage and turnover may be
insufficient to uniquely identify the parameters of even the simplest models (Juston et al., 2010; Luo et al., 2017).
Such problems of parameter 'non-identifiability' or 'equifinality' (Beven, 2006) may introduce considerable
uncertainties into model predictions under changing agro-environmental conditions (e.g. Sierra et al., 2015;
Bradford, 2016; Luo et al., 2017). Making use of the results of this sensitivity and uncertainty analysis, we
calibrated the model against field data obtained from two treatments (bare fallow, animal manure) at the Ultuna



long-term frame trial in Uppsala, Sweden, using measurements of the temporal changes in SOC concentrations
and bulk density and limited data on the soil pore size distribution derived from water retention curves, as well as
surface elevation. As a further test, we also compared predictions of the calibrated model with independent
observations of measurements made in a green manure treatment in the same experiment.

**2 Description of the model**

The model describes the dynamic two-way interactions between SOM storage and turnover and soil porosity and
pore size distribution. A simple conceptual model is adopted to capture how the soil pore space changes as a result
of changes in soil organic matter concentration (Figures 1 and 2). We consider that the total pore volume, $V_p$,
comprises the sum of a constant textural pore volume, $V_{text}$, defined as the minimum value of the pore volume
found in a purely mineral soil matrix without SOM (e.g. Fies and Stengel, 1981; Yoon and Gimenéz, 2012) and a
dynamic structural pore volume comprising both macropores, $V_{mac}$, and an aggregation pore volume, $V_{agg}$. It should
be emphasized here that although the model describes an aggregated pore space generated by microbial turnover
of SOM, soil 'aggregates' themselves are not considered as explicit entities in this model, which instead is based
on the soil pore space. In addition to classifying the soil pore space in terms of its origin, the model also considers
three pore size classes (Figures 1 and 2), with the soil matrix porosity partitioned into mesopores and micropores.
The model currently neglects storage of SOM in macropores because we expect that SOM *per se* would have little
direct influence on the properties of soil macropore networks (e.g. Larsbo et al., 2016; Jarvis et al., 2017), but also
because it would most likely be a minor component of the SOM balance. The pore size distribution in the soil
matrix influences SOM storage and turnover in the model in two ways: firstly, the mineralization rate of SOM in
microporous regions is reduced due to physical protection. Secondly, the partitioning of OM inputs derived from
plant roots between the two pore classes is determined by their relative volumes, in an attempt to mimic in a simple
way how changes in soil structure affect the spatial distribution of root proliferation in soil. SOM is transferred
between the two pore size classes using a simple mixing concept to reflect the homogenizing effects of soil tillage
and faunal bioturbation. In this sense, the model has some conceptual similarities to dual-pore region models used
to quantify the effects of soil structure on water flow and solute transport (e.g. Larsbo et al. 2005).

**2.1 Soil organic matter storage and turnover**

Four pools of organic matter (kg OM m$^{-2}$) are considered in the model, comprising two types (qualities) of organic
matter stored in the two pore regions of the soil matrix (Figures 1 to 3): the model tracks two pools of young
undecomposed organic matter, one stored in parts of the soil in contact with well-aerated mesopore networks and
the other stored in microporous soil regions ($M_{Y(mes)}$ and $M_{Y(mic)}$ respectively). Likewise, the model accounts for
two pools of older microbially-processed organic matter, stored in the mesoporous and microporous regions of
soil respectively ($M_{O(mes)}$ and $M_{O(mic)}$). Both types of organic matter are transferred between the two pore regions
by physical mixing processes, such as tillage and bioturbation. The SOM fluxes and rates of change of storage in
the four pools of organic matter in the model are given by a modified version of the ICBM model (Andrén and
Kätterer, 1997; Wutzler and Reichstein, 2013) extended to account for organic matter storage in two pore regions:

$$\frac{dM_{Y(mes)}}{dt} = I_m + \left(\frac{\phi_{mes}}{\phi_{mes}+\phi_{mic}}\right)I_r - k_Y M_{Y(mes)} + T_Y \tag{1}$$

$$\frac{dM_{O(mes)}}{dt} = \left(\varepsilon\, k_Y M_{Y(mes)}\right) - \left((1-\varepsilon)\, k_O M_{O(mes)}\right) + T_O \tag{2}$$



$$\frac{dM_{Y(mic)}}{dt} = \left(\frac{\phi_{mic}}{\phi_{mes}+\phi_{mic}}\right)I_r - k_Y F_{prot} M_{Y(mic)} - T_Y \tag{3}$$

$$\frac{dM_{O(mic)}}{dt} = \left(\varepsilon\, k_Y F_{prot} M_{Y(mic)}\right) - \left((1-\varepsilon)\, k_O F_{prot} M_{O(mic)}\right) - T_O \tag{4}$$

where $\phi_{mic}$ and $\phi_{mes}$ are micro- and mesoporosity (m$^3$ m$^{-3}$), $k_Y$ and $k_O$ are the first-order rate constants for the

decomposition of fresh and microbially-processed organic matter (year$^{-1}$), $F_{prot}$ is a factor (-) varying from zero to unity that reduces decomposition in the micropore region to reflect physical protection, $\varepsilon$ is an OM retention coefficient (-), $T_Y$ and $T_O$ are source-sink terms (kg m$^{-2}$ year$^{-1}$) for the mixing of organic matter (e.g. by tillage or earthworm bioturbation) between the two pore classes and $I_r$ and $I_m$ are the below-ground (root residues and exudates) and above-ground (litter and organic amendments e.g. manure) inputs of organic matter (kg m$^{-2}$ year$^{-1}$).

It can be seen from equations 1 and 3 that the model assumes that root-derived organic matter is added to the microporous and mesoporous regions in proportion to their volumes, while above-ground litter and organic amendments are added solely to the mesopore region. Finally, the source-sink terms for mass exchange in equations 1 to 4 are given by:

$$T_Y = k_{mix}\left(\frac{M_{Y(mic)}-M_{Y(mes)}}{2}\right) \tag{5}$$

$$T_O = k_{mix}\left(\frac{M_{O(mic)}-M_{O(mes)}}{2}\right) \tag{6}$$

where $k_{mix}$ is the proportion of the stored organic matter which is mixed annually (year$^{-1}$), varying between zero (no mixing) and unity (complete mixing on an annual time scale).

**2.2 Soil physical properties**

The model of SOM turnover and storage described by equations 1 to 6 above considers how the soil pore space

influences SOM dynamics. We now derive a simple model of the feedback effects of SOM on porosity and pore size distribution. Our starting point is the fundamental phase relation for the total soil volume, $V_t$ (m$^3$):

$$V_t = V_s + V_p = V_{s(o)} + V_{s(m)} + V_p = \left\{A_{xs}\left(\frac{M_{s(o)}}{\gamma_o} + \frac{M_{s(m)}}{\gamma_m}\right) + V_p\right\} \tag{7}$$

where $V_s$, $V_{s(o)}$, $V_{s(m)}$ and $V_p$ are the volumes (m$^3$) of solids, organic matter, mineral matter and pore space, $\gamma_o$ and $\gamma_m$ are the densities (kg m$^{-3}$) of organic and mineral matter, $A_{xs}$ is a nominal cross-sectional area in the soil (= 1 m$^2$),

$M_{s(m)}$ is the mass of mineral matter (kg m$^{-2}$) and $M_{s(o)}$ is the total mass of organic matter (kg OM m$^{-2}$) given by:

$$M_{s(o)} = M_{Y(mes)} + M_{O(mes)} + M_{Y(mic)} + M_{O(mic)} \tag{8}$$

The mineral mass, $M_{s(m)}$, in equation 7 is assumed constant and is obtained from user-defined values of a minimum matrix porosity, $\phi_{min}$ (m$^3$ m$^{-3}$), and thickness of the soil layer, $\Delta z_{min}$ (m), corresponding to the minimum soil volume, $V_{t(min)}$ (m$^3$) attained when $M_{s(o)} = 0$:

$$M_{s(m)} = \Delta z_{min}\gamma_m(1 - \phi_{min}) \tag{9}$$

$$V_{t(min)} = A_{xs}\,\Delta z_{min} \tag{10}$$





The volume of organic matter, $V_{s(o)}$, and thus $V_t$, in equation 7 naturally changes as the stored mass of soil organic matter, $M_{s(o)}$, changes. The total soil volume is also affected by changes in the dynamic soil pore volume, which comprises macropores as well as aggregation pore space induced by microbial decomposition of organic matter.

Previous studies suggest that the volume of aggregation pore space, $V_{agg}$, should vary as a linear function of the volume of soil organic matter, $V_{s(o)}$ (e.g. Emerson and McGarry, 2003; Boivin et al., 2009; Johannes et al., 2017). For the sake of simplicity, we also assume here that the soil macroporosity is constant, such that $V_{mac}$ is maintained proportional to the total soil volume. With these assumptions, the total pore volume, $V_p$, is given by:

$$V_p = V_{agg} + V_{text} + V_{mac} = A_{xs} \left\{ f_{agg} \left( \frac{M_{s(o)}}{\gamma_o} \right) + \Delta z_{min} \phi_{min} + \Delta z \, \phi_{mac} \right\}$$ (11)

where $f_{agg}$ is an aggregation factor (m³ pore space m⁻³ organic matter), $\phi_{mac}$ is the macroporosity (m³ m⁻³), $\Delta z$ is the layer thickness (m) and the constant volume of textural pores, $V_{text}$ (m³), is obtained by combining equations 7, 9 and 10 with $M_{s(o)} = 0$. Changes in $V_{s(o)}$ and $V_p$ induce temporal variations in the total soil volume (and therefore layer thickness), porosity and bulk density. Combining equations 7, 9 and 11, gives the soil layer thickness as:

$$\Delta z = \frac{V_t}{A_{xs}} = \frac{\left\{ (1+f_{agg}) \left( \frac{M_{s(o)}}{\gamma_o} \right) \right\} + \Delta z_{min}}{1 - \phi_{mac}}$$ (12)

and the matrix porosity $\phi_{mat}$ (m³ m⁻³), total porosity, $\phi$ (m³ m⁻³), and soil bulk density, $\gamma_b$ (kg m⁻³) as:

$$\phi_{mat} = \frac{V_{agg} + V_{text}}{V_t} = \frac{\left\{ f_{agg} \left( \frac{M_{s(o)}}{\gamma_o} \right) \right\} + \{ \Delta z_{min} \phi_{min} \}}{\Delta z}$$ (13)

$$\phi = \frac{V_{agg} + V_{text} + V_{mac}}{V_t} = \phi_{mat} + \phi_{mac}$$ (14)

$$\gamma_b = \frac{M_{s(o)} + M_{s(m)}}{V_t} = \frac{M_{s(o)} + \left( \Delta z_{min} \gamma_m (1 - \phi_{min}) \right)}{\Delta z}$$ (15)

It is also helpful to derive expressions for porosity and bulk density as functions of the soil organic matter

concentration, $f_{som}$ (kg kg⁻¹), rather than of $M_{s(o)}$, since $f_{som}$ is normally measured in the field. By definition:

$$f_{som} = \frac{M_{s(o)}}{M_{s(o)} + M_{s(m)}}$$ (16)

Combining equations 9 and 16 gives:

$$M_{s(o)} = \frac{f_{som} \Delta z_{min} \, \gamma_m \, (1 - \phi_{min})}{1 - f_{som}}$$ (17)

Substituting equation 17 into equations 13 – 15 and simplifying gives:

$$\phi_{mat} = \frac{\left[ \left\{ \left( \frac{f_{som}}{\gamma_o} \right) f_{agg} + \left( \frac{\phi_{min}(1-f_{som})}{\gamma_m(1-\phi_{min})} \right) \right\} (1 - \phi_{mac}) \right]}{\left\{ \left( \frac{f_{som}}{\gamma_o} \right) (1 + f_{agg}) \right\} + \left( \frac{1 - f_{som}}{\gamma_m(1-\phi_{min})} \right)}$$ (18)

$$\phi = \phi_{mat} + \phi_{mac}$$ (19)

$$\gamma_b = \frac{1 - \phi_{mac}}{\left\{ \left( \frac{f_{som}}{\gamma_o} \right) (1 + f_{agg}) \right\} + \left( \frac{1 - f_{som}}{\gamma_m(1-\phi_{min})} \right)}$$ (20)





In the absence of other governing processes, equations 18 – 20 enable the identification of upper and lower limits of porosity and bulk density that occur at limit SOM concentrations of zero (i.e. a purely mineral soil) and unity

(i.e. organic soils). Setting $f_{som}$ to zero defines the maximum and minimum values of bulk density and porosity as:

$$\gamma_b = \gamma_m (1 - \phi_{min})(1 - \phi_{mac}) \tag{21}$$

$$\phi = \phi_{min} + \phi_{mac} (1 - \phi_{min}) \tag{22}$$

Conversely, bulk density and porosity attain minimum and maximum values respectively in an organic soil where $f_{som} = 1$ kg kg$^{-1}$, such that:

$$\gamma_b = \frac{\gamma_o (1 - \phi_{mac})}{1 + f_{agg}} \tag{23}$$

$$\phi = \left( \frac{f_{agg}}{1 + f_{agg}} \right)(1 - \phi_{min}) + \phi_{mac} \tag{24}$$

The partitioning of the matrix porosity, $\phi_{mat}$, between micro- and mesoporosity is given by:

$$\phi_{mic} = \frac{V_{agg(mic)} + V_{text(mic)}}{V_t} = \frac{\left\{ f_{agg} \left( \frac{\left( M_{Y(mic)} + M_{O(mic)} \right)}{\gamma_o} \right) \right\} + \left\{ F_{text(mic)} \Delta z_{min} \phi_{min} \right\}}{\Delta z} \tag{25}$$

$$\phi_{mes} = \phi_{mat} - \phi_{mic} \tag{26}$$

where $V_{agg(mic)}$ and $V_{text(mic)}$ are the volumes (m$^3$) of aggregation and textural micropores (see Figure 2) and $F_{text(mic)}$ represents the proportion of the textural pore space that comprises micropores. It may be feasible to estimate $F_{text(mic)}$ from data on soil texture, since pore and particle size distributions are similar in the absence of structural pores (e.g. Arya et al., 1999; Yoon and Gimenéz, 2012; Arya and Heitman, 2015).

The model described by equation 20 was first derived by Federer et al. (1993), albeit in a simpler form in which

macroporosity was neglected and $\gamma_o$ and $f_{agg}$ were lumped into one parameter, the bulk density of a purely organic soil given by equation 23 with $\phi_{mac}$=0. They showed that this model could accurately represent the observed relationship between organic matter concentration and bulk density measured on 480 samples of forest soils in north-eastern U.S.A. The validity of this model approach is further confirmed by Figure 4, which shows that equation 20 gives reasonably good fits to paired measurements of bulk density and organic matter concentration

made at three agricultural field sites in Sweden, including the Ultuna frame trial. It should be noted that the composition of OM sources may affect the extent of soil aggregation generated by microbial activity (e.g. Bucka et al., 2019). In this respect, each of the four OM pools could have been characterized by a different aggregation factor. However, we have assumed here that the two qualities of organic matter modify the pore space to the same extent in both the micropore and mesopore regions, so that only a single aggregation factor, $f_{agg}$, is required in the

model. As we will see later, this is because unequivocal parameterization of a more detailed model would be difficult to achieve, given the amount and kinds of data normally available from field experiments. Alternatively, a model of intermediate complexity can be envisaged in which $f_{agg}$ would take different values in micropore and mesopore regions. Such a model would only introduce one additional parameter compared with the simplest case assumed here, but even this modest increase in complexity could cause difficulties with parameter identifiability.




Equations 13, 25 and 26 describe a partitioning of the matrix pore space into two size classes as a dynamic function of soil organic matter storage. This partitioning can also be used to estimate continuous model functions for soil hydraulic properties (water retention, hydraulic conductivity) to enable a straightforward coupling to hydrological models based on Richards' equation. Most commonly used models of soil water retention employ two shape parameters to characterize the pore size distribution. Thus, one requirement of this approach is that one of these two parameters must be assumed to remain constant. We illustrate this approach taking the widely used van Genuchten (1980) equation as an example. If residual water is negligible, the water content $\theta$ ($m^3\,m^{-3}$) is given by:

$$\theta = \phi_{mat}(1 + |\alpha\,\psi|^n)^{\frac{1}{n}-1} \tag{27}$$

where $\psi$ (cm) is the soil water pressure head and $\alpha$ ($cm^{-1}$) and $n$ (-) are shape parameters that reflect the pore size distribution. We assume that $n$ can be held constant, since it is known to be strongly determined by soil texture (e.g. Wösten et al., 2001; Vereecken et al., 2010), while $\alpha$ is allowed to vary, as it is more influenced by the nature of the structural pore space in soil (Assouline and Or, 2013). In this case, $\alpha$ ($cm^{-1}$) is given by:

$$\alpha = \frac{\left[\left(\frac{\phi_{mic}}{\phi_{mat}}\right)^{-\frac{n}{n-1}}-1\right]^{1/n}}{|\psi_{mic/mes}|} \tag{28}$$

where $\psi_{mic/mes}$ is a fixed user-defined pressure head (cm) defining the size of the largest micropore in soil. This model only considers the two pore size classes comprising matrix porosity. However, it is possible to extend this model to account for macropores by making use of dual-porosity concepts (Durner, 1994; Larsbo et al., 2005).

## 3 Application of the model

### 3.1 Sensitivity analysis

We performed a Monte Carlo sensitivity analysis to better understand the behaviour of this new model. We ran 500 simulations with parameter values obtained by Latin hypercube sampling from uniform distributions. The simulations were run for 2000 years to make the outputs independent of the assumed initial conditions. Organic matter was added solely from below-ground residues at a rate (0.02 g $cm^{-2}$ $year^{-1}$) that gave a final organic matter concentration of 0.03 kg $kg^{-1}$ for the mean simulation. The sensitivity of the model parameters was quantified by Spearman rank partial correlation coefficients for three target output variables: the final values of bulk density, $\gamma_b$, soil organic matter concentration, $f_{som}$, and the micropore fraction of the matrix porosity, $f_{mic}$ ($=\phi_{mic}/\phi_{mat}$), as a measure to characterize the soil pore size distribution (see equation 28). Parameter ranges of $F_{prot}$ and $F_{text(mic)}$ ($0.05 < F_{prot} < 0.2$; $0.5 < F_{text(mic)} < 0.9$; see Table 1) were selected to represent a well-structured loamy to fine-textured soil, assuming a maximum pore size of the micropores of 5 µm (i.e. $\psi_{mic/mes} = -600$ cm). Our analysis focuses on matrix pore space properties and SOM, so the macroporosity was fixed at a constant value in these simulations. The sampled ranges for the remaining model parameters shown in Table 1 were selected to approximately match their expected variations based on previous modelling experience.

The partial rank correlation coefficients are shown in Table 1. Not surprisingly, the organic matter concentration $f_{som}$ was most affected by parameters regulating SOM turnover, especially the OM retention coefficient, $\varepsilon$, and the first-order rate coefficient for the microbially-processed OM pool, $k_o$. As expected, the physical protection factor, $F_{prot}$, was also highly significantly (and negatively) correlated with $f_{som}$. Parameters controlling organic matter





turnover also strongly affected the simulated bulk density, $\gamma_b$, along with soil physical parameters, especially the aggregation factor, $f_{agg}$, and the minimum (i.e. textural) porosity, $\phi_{min}$. The pore size distribution, as expressed by the fraction of micropores, $f_{mic}$, was most sensitive to changes in the micropore fraction of the textural pore space, $F_{mic(text)}$ (Table 1). This is encouraging because it is well known that soil texture exerts the most important control

on the pore size distribution in soil. The fraction of micropores was also highly significantly (and negatively) correlated with the mixing coefficient, $k_{mix}$, presumably because this mixing transferred root-derived OM from micropores to mesopores. This is also the reason why the bulk density, $\gamma_b$, and $f_{som}$ are also strongly correlated with $k_{mix}$ (Table 1), given that OM decomposition rates differ between the pore regions.

### 3.2 Parameter identifiability

The fact that model parameters are sensitive does not imply that they will be identifiable in a calibration procedure, since their effects on the target outputs may be correlated (e.g. Luo et al., 2017). We therefore investigated the identifiability of the model parameters using synthetic data generated by 50-year forward simulations of the model for two scenarios with different OM inputs: a bare fallow scenario with no OM inputs and a scenario with a constant OM input of 0.06 g cm$^{-2}$ year$^{-1}$. As initial conditions, the organic matter pools were set to values in

equilibrium with a constant OM input of 0.02 g cm$^{-2}$ year$^{-1}$ giving an initial $f_{som}$ of 0.03 kg kg$^{-1}$. Simulated bulk density, $\gamma_b$, soil organic matter concentration, $f_{som}$, and the soil microporosity, $\phi_{mic}$, were used as target output variables in the calibration. The SOM concentration was assumed to have been sampled every 5$^{th}$ year, while data for bulk density and microporosity were assumed to be available only at the start of the experiment and on two subsequent occasions (after 20 and 50 years). Errors were added to the model simulated values for all three target

output variables to represent measurement and sampling uncertainties due to spatial variability. We calculated these errors assuming 10 replicates per sampling occasion and normally distributed errors with a coefficient of variation of 10 %. The parameter values used to generate the synthetic data are listed in Table 2.

The model was calibrated against the synthetic data using the Powell conjugate gradient method (Powell, 2009) within given parameter ranges defined by minimum and maximum values (Table 2) and using the sum of squared

errors as the goal function. The analysis was repeated 100 times for different initial starting values for the parameters in order to assess the uniqueness of the optimized parameter estimates. Two relatively insensitive parameters, $\gamma_o$ and $\gamma_m$ (Table 1), were assumed to be known and fixed at their true values (Table 2). Two further parameters were excluded from the calibration, namely the aggregation factor, $f_{agg}$, and minimum porosity, $\phi_{min}$. Instead, they were fixed *a priori* by non-linear least squares regression on the synthetic data generated for bulk

density and $f_{som}$ using equation 20 (with $\phi_{mac} = 0$) and known values of $\gamma_o$ and $\gamma_m$ (Table 2). Optimized parameter sets with goal function values less than 10 % larger than the global optimum ($n = 36$) were considered acceptable (Beven, 2006). Figure 5 shows that the best simulation with the calibrated model closely matched the synthetic data for bulk density, SOM and microporosity. Nevertheless, only three of the six parameters ($\varepsilon$, $k_o$ and $F_{mic(text)}$) were identifiable, with values for the 36 best parameter sets limited to narrow ranges around the true values (Figure

6). This was not the case for the three remaining parameters: optimized values of $k_{mix}$ and $k_y$ covered almost the whole tested range, while optimized $F_{prot}$ values were restricted to roughly half of the sampled range (Figure 6).



### 3.3 Model evaluation with data from a long-term field trial

**3.3.1 Field measurements at the Ultuna frame trial**

The model was tested against data from the Ultuna long-term soil organic matter experiment at Uppsala, Sweden (59.82°N, 17.65°E) (Kirchmann et al., 1994; Witter, 1996; Herrmann and Witter, 2008; Kätterer et al., 2011). The climate is cold temperate and sub-humid with an annual mean air temperature of 6.3°C and a mean annual precipitation of 554 mm (1981-2014). The experiment was started in 1956 at the Swedish University of

Agricultural Sciences in order to investigate the long-term effects of mineral N fertilizers and different organic amendments on crop yields, soil organic matter concentrations and soil physical properties. The soil texture in the uppermost 20 cm is clay loam (37% clay, 41% silt and 22% sand).

Of the 15 treatments included in the experiment, three were chosen for model testing: a bare soil treatment (bare fallow) that has received neither mineral N fertilizer nor any organic amendments since the beginning of the

experiment and two other treatments receiving no mineral N fertilizer but 4 t ha$^{-1}$ C as organic amendments every second year in the form of green manure and animal manure, respectively. All three treatments receive P and K fertilizer (20 and 38 kg ha$^{-1}$ yr$^{-1}$) and are annually dug by hand, with the organic amendments mixed into the soil to a depth of 20 cm. The organic amendments were added irregularly at the beginning of the experiment i.e. in 1956, 1960 and 1963, but have since been supplied every second year. Maize has been grown exclusively on all

the cropped plots since 2000. Before 2000, the crop rotation included a sequence of barley, oats, beets (excluded after 1966) and occasionally rape. Samples for the measurement of SOC were taken after harvest of the crops every second year. The three selected treatments show contrasting temporal trends in SOC during the 63 years of the experiment. While SOC concentrations have decreased steadily in the bare fallow treatment, they are still increasing in the plots fertilized with animal manure. Addition of green manure led to a slight increase in SOC

concentrations during the first 10-15 years of the experiment, followed by a period of approximately steady-state conditions and then a slight decline in SOC concentrations on the most recent sampling occasions. Soil bulk density was measured occasionally, i.e. in 1956, 1975 and 1991 (Kirchmann et al. 1994), 1993 (Gerzabek et al., 1997), 1997 (Kirchmann and Gerzabek, 1999), 2009 (Kätterer et al. 2011) and in 2019 (this study). Kätterer et al. (2011) also reported measurements of relative surface elevation in 2009, which we utilize as additional validation data.

Of the three treatments, the bare fallow plots show the largest bulk densities and the animal manure treatments the smallest. Information on the soil pore size distribution was extracted from water retention curves measured on samples taken in the uppermost 10 cm of soil on three different sampling occasions. As soil water retention was not measured at the start of the experiment, we made use of measurements made in 1969 (13 years later) on samples taken from just outside the experimental plots (Wiklert et al., 1983) to initialize the model. Soil water retention

was also measured on four replicate undisturbed core samples taken from the three treatments in 1997, 41 years after the start of the experiment (Kirchmann and Gerzabek, 1999) and on eight replicate samples taken in 2019, although on this occasion only from the animal manure and bare fallow treatments.

### 3.3.2 Parameterization and calibration

The model was simultaneously calibrated against data from the bare fallow and animal manure treatments using

the measurements of average soil bulk density and SOC concentrations in the uppermost 20 cm of soil, as well as the microporosity estimated from soil water retention curves, assuming a value for the maximum pore diameter of micropores of 5 μm (equivalent to a pressure head $\psi_{mic/mes}$ of -600 cm). A factor of 0.5 (Pribyl, 2010) was used to





convert simulated SOM to measured SOC concentrations. We simulated a soil profile consisting of five soil layers, each initially 4.5 cm in thickness. The model was run with an annual time step and a warm-up phase of 5000 years

with constant root-derived OM input to initialize the four SOM pools at a steady-state condition. During the 63-year experimental period, annual average OM inputs from roots and above-ground crop residues were used in the model. Following Kätterer et al. (2011), these were calculated for each treatment from annual yield data and the crop-specific root allocation coefficients reported by Bolinder et al. (2007). The root-derived input of OM to the simulated soil profile was calculated from an assumed root distribution estimated with a Michaelis-Menten-type

function (Kätterer et al., 2011) and distributed uniformly among the soil layers. The organic amendments (8 t OM ha$^{-1}$ every other year in both the animal and green manure treatments) were assumed to be uniformly distributed within the 20 cm depth of soil hand dug by hand. This means that some of this added OM eventually may become incorporated into the subsoil below 20 cm (i.e. the depth of digging), if soil layer thicknesses increase (and bulk density decreases) due to an increase in SOM concentration (see equation 12).

Based on the results of the sensitivity analysis and model calibration against the synthetic data, we decided to calibrate only four parameters, namely the ones that we expected to be clearly identifiable: the input of organic matter during the warm-up period, the fraction of micropores in the textural pore region $F_{mic(text)}$, the OM retention coefficient ε, and the first-order rate coefficient for microbially-processed organic matter, $k_o$ (Table 3). Values for $\phi_{mac}$ and $f_{agg}$ were estimated using equation 20 from non-linear regression between bulk densities and SOM

concentrations assuming a value of $\phi_{min}$ of 0.35 cm$^3$ cm$^{-3}$ (Nimmo, 2013) and including data from all three of the treatments (i.e. bare fallow, animal and green manure; Figure 4). Similarly, van Genuchten´s $n$ was fixed to a value (= 1.073) obtained from a simultaneous fit of equation 28 to the water retention data measured in 2019 in the fallow and animal manure treatments. The remaining parameters were determined *a priori*, because they were less well identified in the calibration against the synthetic data. Given that the micropore region comprises pores

smaller than 5 μm in diameter, we set the physical protection factor $F_{prot}$ to 0.1, a value which lies within the range observed in experiments (e.g. Kravchenko et al., 2015). Following Andrén and Kätterer (1997), we assumed $k_y$ = 0.8 year$^{-1}$. Estimating the mixing coefficient $k_{mix}$ is problematic because it is highly sensitive for all target outputs (table 1) but not identifiable by calibration (Figure 6). From preliminary simulations, we also concluded that $k_{mix}$ must be set to a much smaller value in the warm-up period than during the 63-year experimental period in order

to avoid obtaining unrealistically large calibrated estimates of the OM input prior to the experiment. A smaller $k_{mix}$ value during the warm-up period presumably reflects the crop rotation practiced at the site prior to the experiment, which included frequent grass leys, so that the soil was tilled much less often. For the sake of simplicity, we set $k_{mix}$ to zero during the warm-up period and to 0.05 year$^{-1}$ during the experiment. This gave a calibrated value of the OM input during the warm-up period (0.0064 g cm$^{-2}$ year$^{-1}$; Table 3) that is similar to the root OM input

estimated for the green manure and animal manure plots during the experiment (0.0061 and 0.0071 g cm$^{-2}$ year$^{-1}$ respectively).

The calibration method was the same as described earlier for the synthetic data set. The calibrated model was then applied to the green manure treatment by running a forward simulation using the calibrated parameter values and the treatment-specific OM inputs. Again, a warm-up period of 5000 years was run in order to bring the SOM pools

and total organic matter concentration to an initial steady-state condition. The goodness-of-fit of the model simulations was evaluated by three criteria, i.e. the Pearson correlation coefficient $r$, the root mean squared error RMSE and the mean absolute error MAE (equations 29 to 31). While $r$ is a measure of the strength of the





relationship between the observations and simulations with a value of 1 showing a perfect positive linear relationship and a value of -1 showing a perfect negative linear relationship, RMSE and MAE measure the average
magnitude of the error between observations and simulations. Both of them vary from 0 to ∞ with smaller values representing a better agreement. However, for the RMSE the errors are squared before averaging, which gives comparatively greater weight to larger errors.

$$r = \frac{cov(y,\hat{y})}{\sigma_y \sigma_{\hat{y}}} \tag{29}$$

$$RMSE = \sqrt{\frac{1}{n} \sum_{i=1}^{n} e_i^2} \tag{30}$$

$$MAE = \frac{1}{n} \sum_{i=1}^{n} |e_i| \tag{31}$$

where $y$ and $\hat{y}$ represent the observations and simulation results, respectively, cov is the covariance, $\sigma_y$ ands $\sigma_{\hat{y}}$ are the standard deviations of $y$ and $\hat{y}$, $e$ is the model error, i.e., $y - \hat{y}$, and $n$ is the number of observations. The analyses were carried out with R (version 3.5.1, R Core Team 2018) using the *openxlsx* (Walker, 2019) and *plyr*
(Wickham, 2011) packages.

Figure 7 and Table 4 show that the calibrated model accurately matched the trends observed in soil organic carbon in the bare fallow and animal manure treatments. The data suggests that the soil bulk density increased in the bare fallow treatment during the experiment, whereas it decreased in the animal manure treatment. These trends were also reasonably well described by the model (Figure 7, Table 4). As the soil organic carbon content was accurately
simulated, the somewhat poorer match sometimes found between the model predictions of bulk density and the measurements reflects to a large extent the unexplained variation in the relationship between $\gamma_b$ and $f_{som}$ (equation 20). In this respect, it is likely that the macroporosity, and therefore bulk density, at the time of sampling in autumn may vary from year to year depending on the way the topsoil was dug and the soil conditions at the time of cultivation. Kätterer et al. (2011) found that the elevation of the soil surface in the plots treated with animal manure
was 2.6 cm higher relative to the bare fallow plots in 2009. In comparison, the model predicted a difference in the elevation of the soil surface of 2.7 cm between the two treatments in the same year (2009). The optimized values of the four calibrated parameters (Table 3) are very well constrained and also appear reasonable. The calibrated value of $F_{mic(text)}$ (i.e. the fraction of textural pores smaller than 5 μm) was 0.85 (Table 3). Calculations with the Arya and Heitman (2015) model based on particle size distribution data from the site (Kirchmann et al., 1994)
give a predicted value for $F_{mic(text)}$ of 0.9, which is in excellent agreement with the estimate from model calibration.

Figure 8 shows a comparison of the water retention curves measured in 1997 and 2019 and the corresponding model predictions using equations 27 and 28, alongside the measurements utilized as an initial condition in 1956. The model accurately matched the data in 2019 for both treatments (Figure 8). However, although the shapes of the water retention curves measured in 1997 were also successfully reproduced, the measured matrix porosity
differed significantly between the treatments in 1997 and this difference could not be matched by the model (Figure 8). It is unclear whether this discrepancy can be attributed solely to model error. Spatial variability in the field may also have played a significant role, since only four replicate core samples were taken in 1997. Regardless of the reason for the discrepancy, the results suggest that it should be a reasonable assumption to hold the parameter $n$ in





van Genuchten's (1980) equation constant in dynamic models of soil matrix hydraulic properties. Figure 8 shows

that whilst $n$ is fixed, van Genuchten´s (1980) α increased in the manure treatment, reflecting an improvement in structure, and decreased in the bare fallow, indicating structural degradation. The soil microporosity apparently decreased during the experiment in both treatments, while the mesoporosity remained largely unchanged in the fallow plots and only increased slightly in the manured treatment (Figures 7 and 8). The model simulations suggest some possible explanations for these results, which are surprising at first sight: in the case of the bare fallow plots

with no OM input, we might expect physical protection to lead to a slower decline in the organic matter stock in the micropore region compared with the mesopore region (and thus an increase in the proportion of micropores). However, the bare fallow soil was tilled every year. The simulation results (Figure 9) suggest that this leads to a homogenization of the OM distribution in soil, with a net transfer of OM from the micropore region to the mesopores at a rate that exceeds the difference in decomposition rates between the pore regions. In the case of the

manured plots, the stock of OM in the micropore region decreases in the model as a result of the significant increase in tillage intensity at the onset of the experiment, despite the large increase in the OM input, as the manure is input solely to the mesopore region (Figure 9). Furthermore, a successively smaller proportion of the root OM is added to the micropores as the aggregation mesopore volume increases (equation 3).

### 3.3.3 Model testing using data from the green manure treatment

The model predictions for the green manure treatment tended to underestimate bulk density, whilst clearly overestimating SOC concentrations (Figure 10). The model predicted a steady increase in SOC throughout the experiment, which was not observed in the field. As the animal and green manure treatments only differ slightly in the amount of C provided by roots and straw, the significant difference in SOC concentrations must be related to differences in the quality of the organic amendments. We therefore re-calibrated ε using the data from the green

manure treatment, keeping all other parameters fixed at the values obtained from the calibration against the other two treatments. The resulting calibrated value for ε was 0.14, which significantly improved the fit of the model to the data for both SOC and bulk density (Figure 10, Table 4). The difference in the elevation of the soil surface between the green manure plots and the bare fallow plots measured by Kätterer et al. (2011) in 2009 (= 1.4 cm) was also accurately simulated by the model (= 1.6 cm). The smaller value of ε in the green manure treatment

implies that less of the supplied OM is retained in the soil compared to the organic matter added to the soil as animal manure. This finding is supported by several previous studies that have analyzed data from this experiment with different approaches (e.g. Witter, 1996; Paustian et al., 1992; Hyvönen et al., 1996; Andrén and Kätterer, 1997; Herrmann 2003). Many studies have shown that the quantity and quality of organic amendments can strongly affect SOC turnover rates by altering the biomass, composition and activity of the soil microbial community (e.g.

Blagodatskaya and Kuzyakov, 2008; Dignac et al., 2017). Herrmann et al. (2014) showed that, despite similar levels of microbial activity measured by heat dissipation, the soil from the green manure treatment had a significantly larger $CO_2$ production for the same energy input than the soil from the plots receiving animal manure.

### 4 Conclusions and perspectives

We presented a new model that describes for the first time the dynamic two-way interactions between SOM, soil

pore space structure and soil physical properties. This model should prove useful as a research tool to explore mechanistic understanding of soil structure controls on SOM decomposition and stabilization. With the increasingly widespread application of non-destructive experimental techniques such as microCT tomography, it



seems probable that more data on the mutual interactions of soil structure and SOM will become available that could be used to test and parameterize the model. In this study, we tested the model against data taken from plots with contrasting OM inputs in a long-term field trial at Ultuna, Sweden. In a bare fallow treatment, the bulk density increased and soil profile thickness decreased as the SOC concentration decreased during the experiment, while the opposite trends were observed in plots amended with animal manure. Small changes were also detected during the experiment in the matrix pore size distribution (i.e. the shape of soil water retention curve). Our relatively simple model concept to couple organic matter storage and turnover with soil pore space structure was able to satisfactorily simulate these changes in SOC stocks and soil properties resulting from the contrasting OM inputs.

The model currently neglects some processes that may be important in determining the long-term storage of organic carbon in soil under changing environmental conditions, such as the interactions of organic carbon with mineral phases in soil and the regulation of decomposition rates by both abiotic factors (i.e. soil temperature and moisture) as well as the biomass, community composition and activity of microbial populations (Dignac et al., 2017). Extending the model to account for these processes would be feasible, but it would require more comprehensive data to ensure effective and reliable results from model calibration. The model described here could also be further developed towards a more complete coupled model of soil structure dynamics and soil processes by accounting for the dynamic effects of other physical (e.g. tillage/traffic, swelling/shrinkage) and biological processes (e.g. root growth/decay and faunal activity) on soil pore space properties and OM turnover. It should also be worthwhile to incorporate our model approach into more comprehensive models of the soil-crop system that integrate descriptions of hydrological processes, carbon and nutrient cycling and crop growth. Such a next-generation soil-crop modelling tool should prove useful in supporting a wide range of analyses related to the long-term effects of land use and climate change on SOM dynamics, soil hydrological processes and crop production.

**Acknowledgments**

This work was funded by the Swedish Research Council for Sustainable Development (FORMAS) in the project "*Soil structure and soil degradation: improved model tools to meet sustainable development goals under climate and land use change*" (grant number 2018-02319).

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





**Table 1.** Sampled parameter ranges and Spearman rank partial correlation coefficients (r) between parameters and target outputs. Values marked in bold show a significant correlation (p < 0.01). $f_{som}$ = soil organic matter concentration, $\gamma_b$ = bulk density, $f_{mic}$ = fraction of micropores.

| Parameter | Sampled range | Partial correlation coefficients, r | | |
|---|---|---|---|---|
| | | $f_{som}$ | $\gamma_b$ | $f_{mic}$ |
| 1st order rate coefficient, $k_y$ [year$^{-1}$] | 0.1 – 1.0 | **-0.54** | **0.37** | -0.10 |
| 1st order rate coefficient, $k_o$ [year$^{-1}$] | 0.01 – 0.05 | **-0.82** | **0.70** | **0.32** |
| Physical protection factor, $F_{prot}$ [-] | 0.05 – 0.20 | **-0.46** | **0.28** | -0.08 |
| OM Retention coefficient, $\varepsilon$ [-] | 0.1 – 0.5 | **0.92** | **-0.82** | **-0.30** |
| Mixing coefficient, $k_{mix}$ [year$^{-1}$] | 0 – 0.2 | **-0.68** | **0.50** | **-0.60** |
| Fraction of textural micropores, $f_{text(mic)}$ [-] | 0.5 – 0.9 | **0.24** | -0.16 | **0.96** |
| Density of mineral matter, $\gamma_{min}$ [g cm$^{-3}$] | 2.6 – 2.7 | -0.09 | **0.37** | 0.01 |
| Density of organic matter, $\gamma_{som}$ [g cm$^{-3}$] | 1.1 – 1.4 | -0.03 | **0.33** | -0.01 |
| Minimum porosity, $\phi_{min}$ [cm$^3$ cm$^{-3}$] | 0.3 – 0.4 | **0.162** | **-0.85** | 0.02 |
| Aggregation factor, $f_{agg}$ [-] | 2 – 4 | 0.0 | **-0.50** | 0.02 |






**Table 2. Parameter values used to generate the synthetic data and the sampled range in the model calibration.**

| Parameters | Value used for data generation (true value) | Sampled range during calibration |
|---|---|---|
| 1st order rate coefficient, $k_y$ [year$^{-1}$] | 0.40 | 0.1 – 1.0 |
| 1st order rate coefficient, $k_o$ [year$^{-1}$] | 0.02 | 0.005 – 0.1 |
| Mixing coefficient, $k_{mix}$ [year$^{-1}$] | 0.05 | 0 – 0.3 |
| Microbial efficiency, $\varepsilon$ [-] | 0.3 | 0.1 – 0.6 |
| Physical protection factor, $F_{prot}$ [-] | 0.3 | 0.05 – 1.0 |
| Fraction of textural micropores, $F_{text(mic)}$ [-] | 0.5 | 0.2 – 0.8 |
| Density of mineral matter, $\gamma_{min}$ [g cm$^{-3}$] | 2.7 | |
| Density of organic matter, $\gamma_{som}$ [g cm$^{-3}$] | 1.2 | |
| Minimum layer thickness, $\Delta z_{(min)}$ [cm] | 16 | |
| Minimum porosity, $\phi_{min}$ [cm$^3$ cm$^{-3}$] | 0.4[a]/0.41[b] | |
| Aggregation factor, $f_{agg}$ [-] | 5.0[a]/4.92[b] | |

[a] used for data generation, [b] estimated by regression (Figure 4) and fixed during calibration



**Table 3. Fixed parameters and range of parameter values included in the calibration, as well as the final parameter
estimates after calibration. The range of the best-fit parameter values for the calibration runs with goal function values
no more than 5% larger than the value for the best simulation ($n = 85$) is given within parenthesis.**


| Parameters | Fixed value | Sampled range | Calibrated value |
|---|---|---|---|
| 1st order rate coefficient, $k_y$ [year$^{-1}$] | 0.80[a] | | |
| 1st order rate coefficient, $k_o$ [year$^{-1}$] | | 0.01 – 0.1 | 0.036 (0.031 – 0.039) |
| Mixing coefficient, $k_{mix}$ [year$^{-1}$] | 0.05 | | |
| OM Retention coefficient, $\varepsilon$ [-] | | 0.2 – 0.7 | 0.37 (0.35 – 0.39) |
| Physical protection factor, $F_{prot}$ [-] | 0.1[b] | | |
| Fraction of textural micropores, $F_{text(mic)}$ | | 0.5 – 0.9 | 0.85 (0.84 – 0.87) |
| Density of mineral matter, $\gamma_{min}$ [g cm$^{-3}$] | 2.7 | | |
| Density of organic matter, $\gamma_{som}$ [g cm$^{-3}$] | 1.2 | | |
| Minimum layer thickness, $\Delta z_{(min)}$ [cm] | 4 | | |
| Minimum porosity, $\phi_{min}$ [cm$^3$ cm$^{-3}$] | 0.35[c] | | |
| Macroporosity, $\phi_{mac}$ [cm$^3$ cm$^{-3}$] | 0.152[d] | | |
| Aggregation factor, $f_{agg}$ [-] | 2.46[d] | | |
| OM input warm-up [g cm$^{-2}$ year$^{-1}$] | | 0.005 – 0.009 | 0.0064 (0.0061 – 0.0066) |

[a] Andrén and Kätterer (1997), [b] Kravchenko et al. (2015), [c] Nimmo (2013), [d] Figure 4





**Table 4. Goodness of fit of the model simulations to observed bulk density and soil organic carbon concentration. r =**
**correlation coefficient. RMSE = root mean squared error. MAE = mean absolute error.**

| | Parameter | r | RMSE | MAE |
|---|---|---|---|---|
| | | **Fallow** | | |
| Calibration | Bulk density [g cm⁻³] | -0.20 | 0.05 | 0.04 |
| | Soil organic carbon [kg kg⁻¹] | 0.95 | 0.0005 | 0.0004 |
| | | **Animal manure** | | |
| | Bulk density [g cm⁻³] | 0.99 | 0.04 | 0.04 |
| | Soil organic carbon [kg kg⁻¹] | 0.89 | 0.0009 | 0.0007 |
| | | **Green manure ($\varepsilon = 0.37$)** | | |
| Validation | Bulk density [g cm⁻³] | 0.94 | 0.08 | 0.07 |
| | Soil organic carbon [kg kg⁻¹] | 0.04 | 0.004 | 0.004 |
| | | **Green manure ($\varepsilon = 0.14$)** | | |
| | Bulk density [g cm⁻³] | 0.98 | 0.06 | 0.05 |
| | Soil organic carbon [kg kg⁻¹] | 0.37 | 0.0008 | 0.0007 |





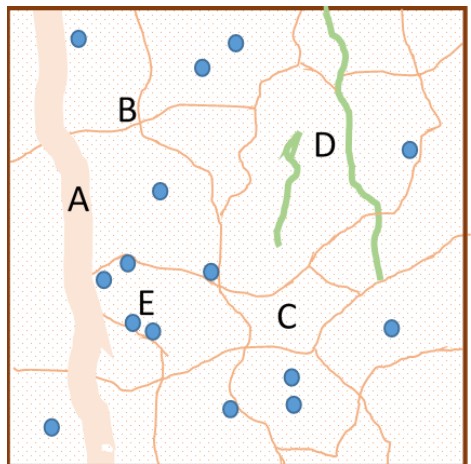

Figure 1. Schematic illustration of the conceptual model with the soil pore space comprising macropores (A), mesopores (thin lines, B) and micropores (dotted regions, C) and with two qualities of organic matter: particulate organic matter (POM e.g. decaying roots; green lines, D), and microbially-processed organic matter (blue circles, E), both of which are stored either in contact only with micropores (and therefore partially protected from decomposition) or in contact with mesopores.





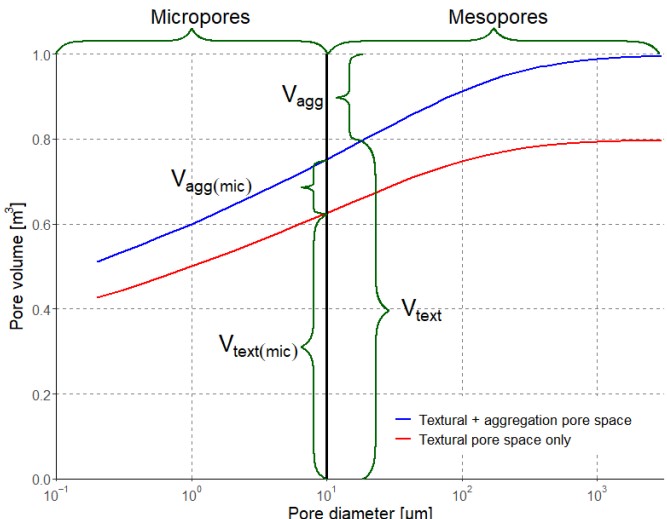


Figure 2. Schematic illustration of pore volumes and pore classes in the model (for explanation of symbols see text). In this example, macroporosity has been neglected and the total pore space is comprised of 80 % textural pores and 20 % aggregation pores induced by soil organic matter, with a maximum micropore diameter of 10 µm.





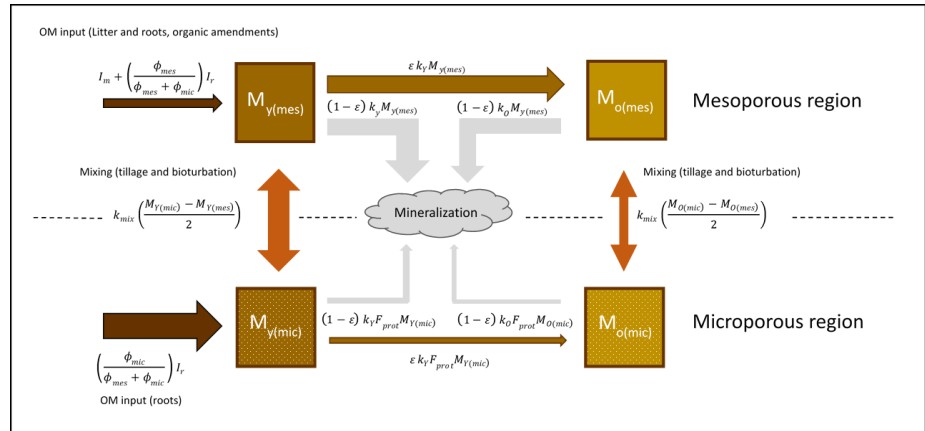

Figure 3. Schematic diagram of the structure of the organic matter model showing storages and flows. For explanations of symbols see the text in connection with equations (1) to (6).

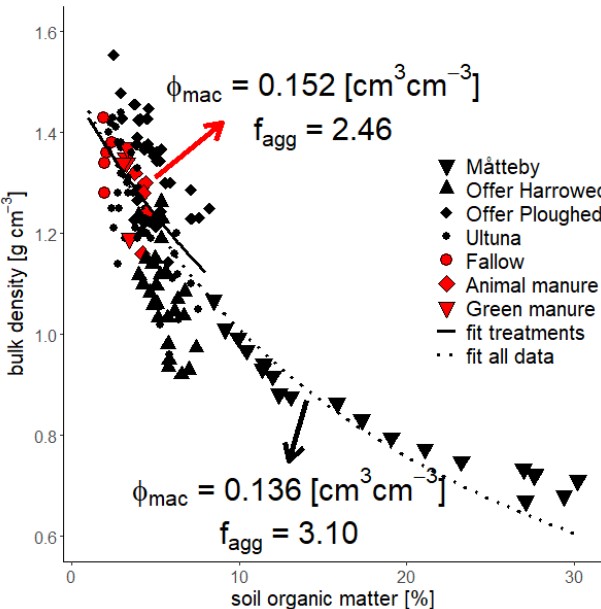

Figure 4. Equation 20 fitted to data from three Swedish field sites (Ultuna data taken from Kirchmann et al., 1994, Gerzabek et al., 1997, Kirchmann and Gerzabek, 1999 and Kätterer et al., 2011; Måtteby data taken from Larsbo et al., 2016, with the soil under grass; Offer data taken from Jarvis et al., 2017; 'harrowed' soil had been ploughed and harrowed (samples were taken at 2-6 cm depth), 'ploughed' soil was only ploughed (samples were taken at 13-17 cm depth). Data used in this study is highlighted in red (fallow, animal manure and green manure). Soil organic matter content was estimated from soil organic carbon by multiplying by 2 (Pribyl, 2010). Equation 20 was fitted by non-linear least-squares regression assuming 'a priori' that $\gamma_m = 2.7$ g cm$^{-3}$, $\gamma_o = 1.2$ g cm$^{-3}$ and $\emptyset_{min} = 0.35$ cm$^3$ cm$^{-3}$.



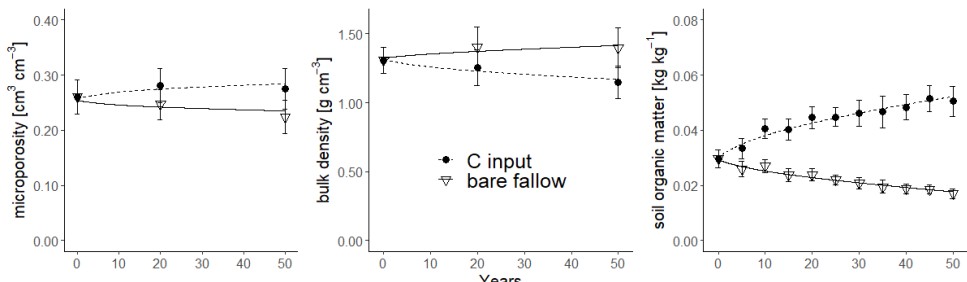

Figure 5. Synthetic data (symbols; bars show standard deviations) for microporosity, bulk density and soil organic matter concentration and model simulations (lines) after calibration.



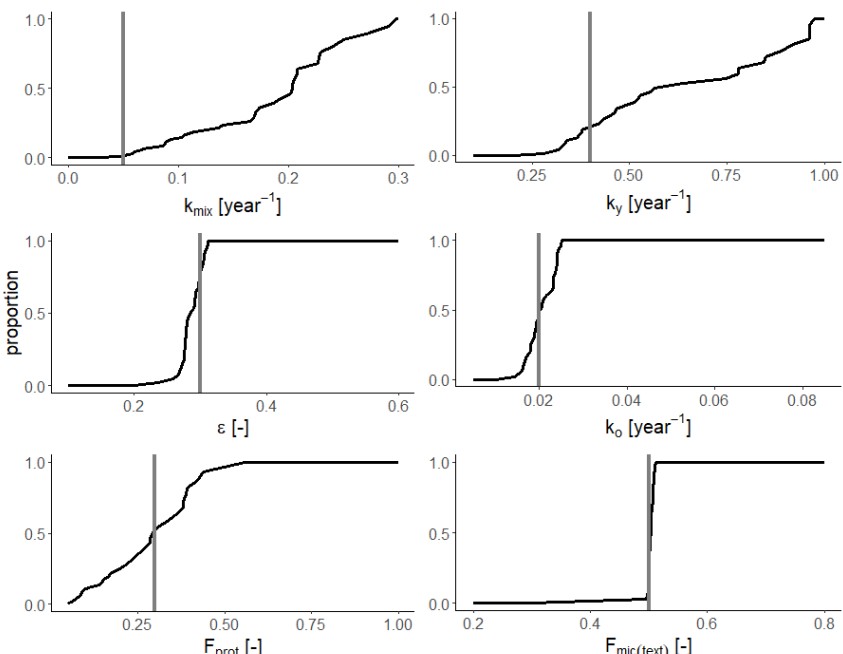

Figure 6. Cumulative frequency distributions of parameter estimates for the 36 best parameter sets of 100 calibration runs against synthetic data for soil bulk density, SOC and microporosity. The grey lines mark the true values used to generate the synthetic data.



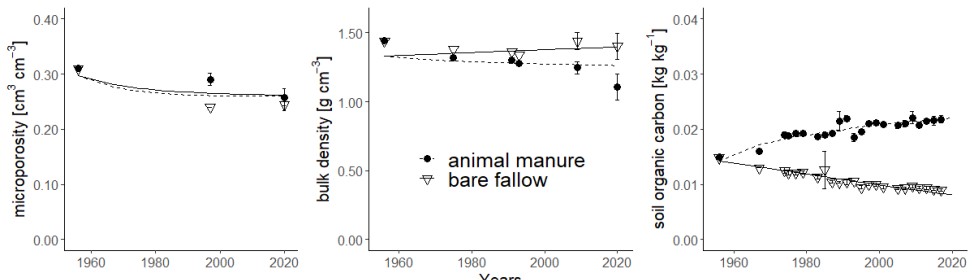

Figure 7. Observed (symbols; bars show standard deviations) and simulated (lines) microporosity [cm$^3$ cm$^{-3}$], bulk density [g cm$^{-3}$] and soil organic carbon concentration [kg kg$^{-1}$] for the fallow and animal manure treatments.





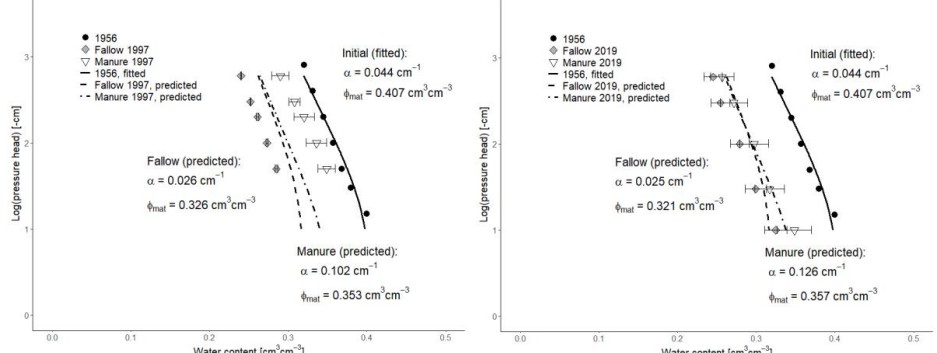

Figure 8. Observed (symbols; bars show standard deviations) and simulated (dashed and dotted lines) soil water retention curves in the fallow and animal manure treatments using equations 27 and 28. The measurements used as the initial condition in 1956 are also shown, together with a fitted curve. Van Genuchten's *n* was fixed at 1.073 for all water retention curves.


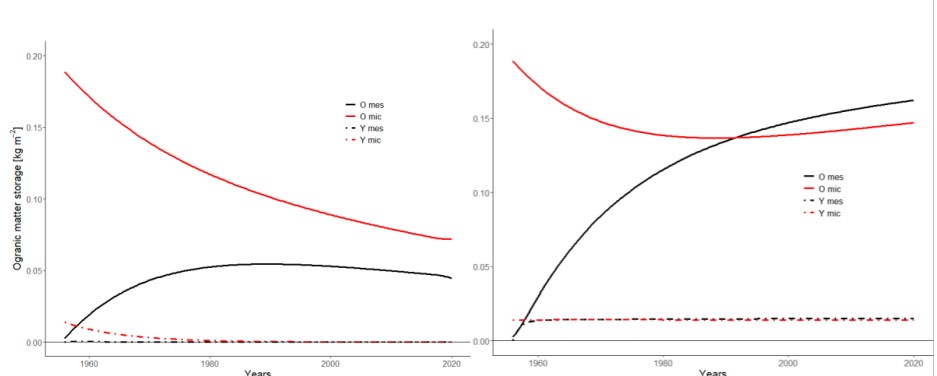

Figure 9. Simulated temporal development of young undecomposed (Y) and older microbially-processed (O) organic matter [kg m$^{-2}$] stored in meso- and microporous regions in the bare fallow (left) and manure (right) treatment.



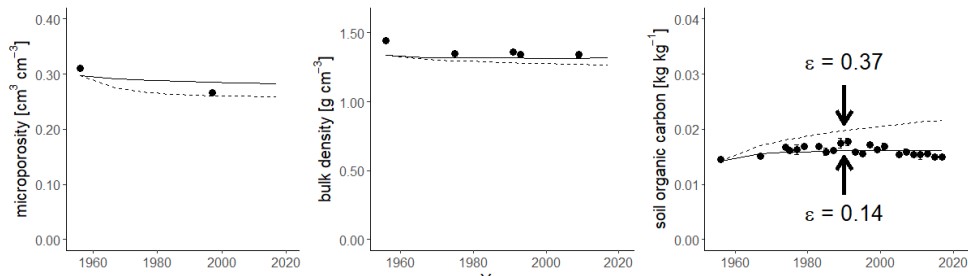

Figure 10. Observed (symbols; bars show standard deviations) and simulated (lines) microporosity [cm$^3$ cm$^{-3}$], bulk density [g cm$^{-3}$] and soil organic carbon concentration [kg kg$^{-1}$] for the green manure treatment for two different values of the OM retention coefficient, ε.