# Peer review of "Modelling dynamic interactions between soil structure and the storage and turnover of soil organic matter"

_Biogeosciences, 2020_

## Referee Comment (RC1) · Thomas Wutzler (Referee) · 19 May 2020

Summary

The study contributes a new model on the dynamical between feedback soil organic matter (SOM) decomposition and soil aggregate structure. Like other models it employs the concept that the addition of low-density organic matter modifies both, the soil layer thickness, porosity, and the bulk density, but is the first study to my knowledge to explicitly discuss this feedback. It explicitly models retardation of of SOM decomposition by aggregation and associated micropores. The approach is demonstrated using a simple parsimonious SOM model at pedon scale with a sensitivity analysis and

a model calibration to a long-term field study. It will be a welcome contribution to the SOM modeling community. I enjoyed reading the manuscript. It is well written and the logical flow is clear to me. The study could be made stronger by including a simulation/calibration without the feedback and comparing the improvements between the two versions.

General comments

I missed a discussion on implications and results on whether the presented feedback is important for understanding or prediction of SOM dynamics or model structure. The authors showed that the relatively simple model could already predict differences in SOM and soil structure by different inputs. However, to what extent could this also be modeled without SOM influencing the soil structure? Although the paper holds enough new insights to be published, I encourage the authors to take the extra work to compare to a model version where the feedback is switched off. For example by calibrating time-constant bulk densities and parameters to the three input-scenarios.

The conclusions currently read more like a discussion. They could be sharpened to what readers should "take home" for their work from this study. What are the most important parameters and feedbacks that you think they need to consider in their experiments and studies?

There are already models that let SOM decomposition affect soil structure. For example in the model of Ahrens et al. 2015 (see also Yu 2020 eq. S28a) SOM dynamics affects bulk soil density and soil volume and this in turn affects modeled concentrations, changes in soil volume, and transport processes. They applied the same concept of Federer 1993 as in the current manuscript, but incorporated many more processes so that this feedback was not explicitly discussed. The present manuscript additionally partitions micro- and mesoporosity and models protection by aggregation. A little comparison in the discussion or introduction would be nice.

P4L103: The authors argue that macropores probably are only a minor balance of
SOM balance. Contrary, some researchers think, that macropores are a hot spot of SOM turnover and together with the rhizosphere are the most important places to study. Especially for systems with active earth worms this has been shown (e.g. Don et al. 2008).

eq. 7 and 11 seem to both add volume and additional pore space with addition of OM. In an alternative mind model putting dissolved organic matter or root exudates into soil would partly fill up existing pores. Please, add some explanation of assumptions to this part.

I miss a paragraph how the model was integrated in time. I assume an explicit time (Euler forward) step much lower than the 5 years of distance between observations. How did you track the changes in soil depth (eq. 12) in the comparison to data?

Minor comments.

The discussion at p3L55 argues about soil structure affecting SOM dynamics. If one could show that it is not only affecting fast pools, then this argument could be made even stronger to affecting SOM stocks and soil carbon sequestration.

The font sizes in the figures are often very small, which makes it difficult to read the print-version.

eq.5 and 6: Why is there a factor of 1/2?

Please, check consistency of mathematical symbols. E.g. delta.z_min is sometimes written with min as subscript and sometimes with parenthesis (Table 1) denoting density gamma_o and gamma_m or gamma_org and gamma_min. F_text_mic or F_mic_text (fig. 6).

p6L165: Parameter f_agg is introduced here. To my reading its quite an important parameter. I recommend explaining it (here or somewhere) in more detail. Does it correspond to the porosity of the volume occupied by organic matter?

[Figure]

eq 21-24: please, use a different symbols at the left hand side than in (19) and (20) to denote the quantities to use assumption of f_som = 0 r f_som = 1.

Sect 3.2: Given the 5 years interval of SOM measurements the non-identifyability of the fast turnover pool is expected. Could you think of additional observations or sub-experiments that could inform the shorter time scale?

Sect 3.2. The mixing ratio was quite influential in Table 1. I assume in the identifyability analysis it correlated strongly with other parameters - which ones? Could this lead to potential model simplifications?

P11L335: "root litter input was distributed uniformly across depth". What do you expect to be the effect of distribution root litter input with an exponentially decreasing profile? How do you treat partitioning of given total root input to the modeled top soil and the non-modeled lower depth?

Fig 1: The dotted regions were not visible in my printout. Please adapt the pattern.

Fig 2: The placing of the braces confused me. Vor micropores its at the maximum pore diameter for mesopores the lower boundary of the upper brace coincides with the blue line. To my understanding it should instead coincide with he red line at the upper diameter.

Fig 3: Cannot read the subscripts in this figure. Please, adjust the font sizes. (Also in the other figures)

Fig 8: I had to search for the difference between left and right panel. Please describe in the legend or make the font of the years 1997 or 2019 more prominent.

Fig 9: Figure headings (bare fallow, manure) in addition to the legent would help the reader.

References

Ahrens B, Braakhekke M, Guggenberger G, Schrumpf M & Reichstein M (2015)

Contribution of sorption, DOC transport and microbial interactions to the 14C age of a soil organic carbon profile: Insights from a calibrated process model.Soil Biology and Biochemistry, Elsevier BV, 88 , 390-402 10.1016/j.soilbio.2015.06.008 suppl: https://ars.els-cdn.com/content/image/1-s2.0-S0038071715002138-mmc1.pdf

Yu L, Ahrens B, Wutzler T, Schrumpf M & Zaehle S(2019)Jena Soil Model: a microbial soil organic carbon model integrated with nitrogen and phosphorus processes.Copernicus GmbH, 10.5194/gmd-2019-187

Don A, Steinberg B, Schöning I, Pritsch K, Joschko M, Gleixner G & Schulze E(2008)Organic carbon sequestration in earthworm burrows.Soil Biology and Biochemistry, Elsevier BV, 40 , 1803-1812 10.1016/j.soilbio.2008.03.003
* * *

---

## Referee Comment (RC2) · Anonymous Referee #2 · 11 Jun 2020

In this work, the authors propose a new framework to model soil organic matter turnover, which includes a two-way coupling between SOM storage and soil porosity. The model considers four pools of organic matter, with the dynamics described by four coupled differential equations. The novelty consists in using additional pools to divide the organic matter between micropore and mesopore soil regions, each one characterized by its own fluxes and decomposition rates. In my opinion such a model indeed can bring new insights about the dynamical feedback between soil physical properties and SOM decomposition, and can be an important contribution to the field. Although I find the paper interesting, I have some concerns. In particular I would have appreciated a more detailed discussion of the advantages of this new model.

[Figure]

My recommendation is publication of this manuscript subject to a revision based on comments listed below.

1 - I find that the paper is in general well written, but the section with the description of the model is very confusing and needs to be improved. I would suggest to first write the full model including the feedback on porosity, and only afterward to follow with all the necessary derivations. Also, it is not clear by looking at the equations which parameters are kept constant, one has always to search in the text. One solution is to use upper case for functions and lower case for constants. Please also double check the notation, for example the density of mineral matter is $\gamma_m$ on pg.6 and $\gamma_{min}$ in all tables.

2 - The abstract states that the model successfully reproduces the soil water retention curves. I find this statement too strong due to the discrepancy of the curves for the year 1997.

3 - I would like to see an extended discussion on the $k\_mix$ and $F\_prot$, since these parameters are at the core of the discussed feedback. For large values of $k\_mix$ and $F\_prot \sim 1$ the soil structure properties have to become less important to the dynamics of SOM turnover. Could the authors comment on this transition to the regime where the soil porosity becomes less relevant for the model outcome? I would also appreciate a short comment on the choice of the sampled range for the sensitivity analysis (and also the choices for calibration).

―――――――――――――――――

---

## Author Comment (AC1) · 16 Jun 2020

We would like to thank Dr. Wutzler for his kind comments and for his valuable and constructive suggestions for improving the paper. We will provide responses to all the comments at a later stage, but here we would like to say a few words about the extremely interesting question he raised concerning a comparison of our model with a simpler model that does not account for the two-way interactions of soil organic matter (OM) with soil structure.

The simple two-pool ICBM model is obtained if the interactions between organic matter and soil structure are removed from our model. In principle, for the same parameterization, the predictions of our model must diverge from ICBM for two or more treatments with contrasting OM input rates. This is because ICBM is strictly a first-order kinetic model, such that steady-state OM contents are a linear function of the OM input. In contrast, our extended model, which incorporates soil structure-OM interactions, does not show an exact linear response to OM inputs, and this non-linearity becomes more marked as the mixing between the pore regions becomes weaker. Nevertheless, successful applications of the ICBM model to the data from the Ultuna frame trial have already been published by Juston et al. (Ecological Modelling, 221, 1880-1888) for data available until 2007 and by Poeplau et al. (Geoderma 237/238, 246-255) for data until 2013. This paradox may be a consequence of the fact that for the duration of the frame trial, the departure from first-order behavior is not so apparent and is overshadowed by noise in the data. We will test how well ICBM can be calibrated to the extended dataset now available until 2019 and we will report these results in the revised paper. However, even if a simpler OM model such as ICBM can be calibrated satisfactorily to time-series of OM measurements at one site, our model that explicitly incorporates soil structure-OM feedbacks has many important advantages. This is because it enables simulations of the effects of soil structure and physical protection on OM turnover in contrasting soil types (e.g. sand vs. clay) explicitly and directly from measured particle size distributions, without having to resort to re-calibrating model parameters describing OM turnover for each soil, as was done, for example, by Poeplau et al. (Geoderma 237/238, 246-255). In principle, our model also has a much broader range of potential management applications. For example, it could be used to simulate the effects of contrasting tillage systems on SOC dynamics, as well as the effects of faunal bioturbation on OM stabilization.

We would also like to emphasize here that in discussing the importance of accounting for soil structure effects on SOM storage in simulation models, we should not ignore "the other side of the coin", namely the importance of SOM for soil structure. We feel that the inclusion in our model of the effects of SOM on porosity, pore size distribution and soil water retention, is a very important advance compared to other models,

because it enables straightforward links to models of soil hydrology, plant growth and therefore OM inputs to soil. This kind of dynamic soil-plant model would encompass, for the first time, a complete description of all the physical feedback mechanisms determining organic C sequestration in soil. We will expand our discussion of these important issues in the revised version of the paper.

———————————————————

---

## Author Comment (AC2) · 16 Jun 2020

We would like to thank the referee for the perceptive comments and constructive suggestions for improving the paper. We will provide responses to all the comments at a later stage, but here we would like to write a few words about one of the points raised by the referee, namely the need to explain in more detail the advantages of our new model. We agree that a detailed discussion of the advantages of modelling interactions between soil structure and soil organic matter storage and turnover is warranted. We did write about this topic in the introduction, but only in rather general terms. We propose to include a more detailed summary of how we see the particular advantages of

our modelling approach in the "Conclusions and perspectives" section. We did mention some of these advantages in our response to Thomas Wutzler (referee 1), which we can briefly repeat here. Our model will enable investigations of the effects of contrasting soil textures, tillage systems, and faunal bioturbation on soil organic matter turnover and storage. With respect to "the other side of the coin" (the effects of organic matter on structure), our model predicts changes in porosity and soil water retention, which enables straightforward links to models of soil hydrology and crop growth. This opens up the possibility to develop a soil-plant model that would encompass, for the first time, a complete dynamic description of all the physical feedback mechanisms in the carbon and water cycles that determine soil quality and organic carbon sequestration in soil.

―――――――――――――――――――

---

## Referee Comment (RC3) · Anonymous Referee #3 · 3 Jul 2020

Meurer and colleagues describe a modified version of the ICBM model which is intended to describe a feedback between SOC formation and decomposition and its effects on bulk density and pore size distribution. While the premise of the study is very interesting it falls short in proving that the feedback between micropore space and SOC decomposition is needed to describe SOC dynamics.

I would ask the authors to clarify and work on the following points:

- Please do a more thorough literature research: Before Federer et al. (1993) a couple authors have used equations similar to the Federer one, maybe even your Equation 20 (Adams, 1973; Rawls, 1983). These are just two examples - probably you can work

your way backwards from here. Tranter et al. (2007) provide a good overview of the literature and show how soil texture affect mineral soil bulk density.

- It would be interesting for the reader to see how much of bulk density changes is due to the difference in density between minerals and soil organic matter (mass effect), and how much due to changes in porosity (difference in porosity between minerals and soil organic matter?). Figure 7 suggests that the microporosity effect is minimal and the increase/decrease in bulk density is solely driven by the decrease/increase in SOC. Please provide some numbers how important SOC changes are for changes in microporosity.

- You set F_prot a priori based on literature values. I think you have to provide more background to the reader how they were derived. SOC is then decomposing at a speed of 10 percent in micropores. Is this well constrained by experiments?

- You use the term 'warm-up'. Please correct to spin-up.

- Please provide a complete list with all symbols and abbreviations. The reader can get lost in the amount of equations otherwise.

References Adams, W., 1973. The effect of organic matter on the bulk and true densities of some uncultivated podzolic soils. Journal of Soil Science 24, 10-17.

Federer, C.A., Turcotte, D.E., Smith, C.T., 1993. The organic fraction–bulk density relationship and the expression of nutrient content in forest soils. Canadian Journal of Forest Research 23, 1026-1032.

Rawls, W.J., 1983. ESTIMATING SOIL BULK DENSITY FROM PARTICLE SIZE ANALYSIS AND ORGANIC MATTER CONTENT1. Soil Science 135, 123-125.

Tranter, G., Minasny, B., McBratney, A.B., Murphy, B., McKenzie, N.J., Grundy, M., Brough, D., 2007. Building and testing conceptual and empirical models for predicting soil bulk density. Soil Use and Management 23, 437-443.

---

## Author Comment (AC5) · 10 Jul 2020

Meurer and colleagues describe a modïfied version of the ICBM model which is intended to describe a feedback between SOC formation and decomposition and its effects on bulk density and pore size distribution. While the premise of the study is very interesting it falls short in proving that the feedback between micropore space and SOC decomposition is needed to describe SOC dynamics.

Authors' response: We would like to thank the referee for his valuable feedback. The paper will be improved as a result.

[Figure]

I would ask the authors to clarify and work on the following points: - Please do a more thorough literature research: Before Federer et al. (1993) a couple authors have used equations similar to the Federer one, maybe even your Equation 20 (Adams, 1973; Rawls, 1983). These are just two examples - probably you can work your way backwards from here. Tranter et al. (2007) provide a good overview of the literature and show how soil texture affect mineral soil bulk density.

Authors′ response: Thank you. Yes, Federer et al. were not the first to apply the model. We will this relevant literature in the text where currently we only mention Federer et al..

- It would be interesting for the reader to see how much of bulk density changes is due to the difference in density between minerals and soil organic matter (mass effect), and how much due to changes in porosity (difference in porosity between minerals and soil organic matter?).

Authors′ response: Yes, this is an interesting question. We interpret the second part of this sentence ("changes in porosity") to mean the effects of aggregation. This question can be answered by analyzing equation 20. We will add a figure to the paper based on this equation showing how the relationship between bulk density and organic matter concentration varies with different values of fagg (the aggregation factor). We attach the proposed figure here. The curve for fagg=0 (i.e. no aggregation) shows that the different densities of organic and mineral matter have only a minor effect. Aggregation dominates the effects of organic matter on bulk density. We can mention here that that the simpler version of this bulk density model previously published does not allow for this kind of analysis, since it does not distinguish between these two effects.

- Figure 7 suggests that the microporosity effect is minimal and the increase/decrease in bulk density is solely driven by the decrease/increase in SOC.

Authors′ response: In this model application, all the parameters in equation 20 (for calculating bulk density) are considered as constants except for the organic matter

content, so yes, the increase/decrease in bulk density is indeed solely driven by the decrease/increase in SOM.

- Please provide some numbers how important SOC changes are for changes in microporosity.

Authors' response: Equation 25 shows how the time-varying SOC content and the (constant) soil textural pore space affect microporosity. The results of the sensitivity analysis suggest that the balance between microporosity and mesoporosity is most strongly determined by soil texture, which certainly agrees with past empirical experience. This was already discussed at lines 252 – 255.

- You set F_prot a priori based on literature values. I think you have to provide more background to the reader how they were derived. SOC is then decomposing at a speed of 10 percent in micropores. Is this well constrained by experiments?

Authors' response: This value is based on a study published by Kravchenko et al. (2015) in which they used X-ray tomography to show that the decomposition rates of intra-aggregate particulate SOM were 3 – 15 times faster in the presence of connected networks of aerated soil pores > 13 $\mu$m in diameter than in the absence of such pores (see Introduction ll. 55 – 57). We chose the value for Fprot based on this range. This was mentioned in the text at lines 349 – 351 and also in Table 3. Clearly, more experiments of this kind will help to better constrain this parameter value in the future. We discussed this at lines 441 – 444.

- You use the term 'warm-up'. Please correct to spin-up.

Authors' response: We will change "warm-up" to "spin-up" in the revised version of the manuscript.

-Please provide a complete list with all symbols and abbreviations. The reader can get lost in the amount of equations otherwise.

Authors' response: We will add a list with symbols and abbreviations.

[Figure]

[Figure]

[Figure]

**Fig. 1.** Effect of fagg

---

## Author Response (AR1)

[revised manuscript text omitted]

**Responses to the Editor**

*Please, carefully revise your manuscript, specifically with regard to the follwing points:*

*(i) please, include a detailed discussion of the advantages of the model as pointed out by both revieer 1 and 2,*

Response: Yes, we have done so. We extended the discussion of the advantages of our model in the final section of the paper.

*(ii) I agree with reviewer 2 that the model description can be improved. The model description should also be accessible to modellers outside the immediate area of reserach and other scientists interested in SOM turnover.*

Response: The model description now uses sub-headings to structure the text in a better way. We have also improved the clarity of the explanations and given a much fuller discussion of the aggregation factor, as requested by referee #1, including an illustrative new figure. This figure also enabled us to answer one of the interesting questions raised by referee #3. We have also added a new table containing a list of all variables and symbols used in the model description (supplementary material, Table S1).

We now feel very confident that the model description should be accessible and understandable for all researchers in the biogeosciences.

*(iii) as reviewer 3 pointed out, the older literature must be better included in your manuscript.*

Response: Yes, we have now included the older literature in the manuscript.

**Responses to referee #1**

*The study contributes a new model on the dynamical between feedback soil organic matter (SOM) decomposition and soil aggregate structure. Like other models it employs the concept that the addition of low-density organic matter modifies both, the soil layer thickness, porosity, and the bulk density, but is the first study to my knowledge to explicitly discuss this feedback. It explicitly models retardation of SOM decomposition by aggregation and associated micropores. The approach is demonstrated using a simple parsimonious SOM model at pedon scale with a sensitivity analysis and a model calibration to a long-term field study. It will be a welcome contribution to the SOM modeling community. I enjoyed reading the manuscript. It is well written and the logical flow is clear to me.*

Response: We would like to thank Dr. Wutzler for his kind comments and for his constructive suggestions for improving the paper.

*The study could be made stronger by including a simulation/calibration without the feedback and comparing the improvements between the two versions.*

Response: See our response to the next comment below

*General comments*

*I missed a discussion on implications and results on whether the presented feedback is important for understanding or prediction of SOM dynamics or model structure. The authors showed that the relatively simple model could already predict differences in SOM and soil structure by different inputs. However, to what extent could this also be modeled without SOM influencing the soil structure? Although the paper holds enough new insights to be published, I encourage the authors to take the extra work to compare to a model version where the feedback is switched off. For example by calibrating time-constant bulk densities and parameters to the three input-scenarios.*

Response: We have now run ICBM against the SOC data for the manure and bare fallow treatments and it performs almost as well as the model described in our paper (RMSE's are slightly larger than those shown in Table 5), albeit with different parameter values: the retention efficiency is similar (0.35 vs. 0.37) but $k_o$ is much smaller (0.015 vs. 0.036 year$^{-1}$), since physical protection is not modelled explicitly. However, even if a simpler OM model such as ICBM can be calibrated reasonably well to time-series of OM measurements at one site, our model that explicitly incorporates soil structure-OM feedbacks has many important advantages. This is because it enables simulations of the effects of soil structure and physical protection on OM turnover in contrasting soil types (e.g. sand vs. clay) explicitly and directly from measured particle size distributions, without having to resort to re-calibrating model parameters describing OM turnover for each soil, as was done, for example, by Poeplau et al. (Geoderma 237/238, 246-255). In principle, our model also has a much broader range of potential management applications. For example, it could be used to simulate the effects of contrasting tillage systems on SOC dynamics, as well as the effects of faunal bioturbation on OM stabilization. We would also like to emphasize here that in discussing the importance of accounting for soil structure effects on SOM storage in simulation models, we should not ignore "the other side of the coin", namely the importance of SOM for soil structure. We feel that the inclusion in our model of the effects of SOM on porosity, pore size distribution and soil water retention, is a very important advance compared to other models, because it enables straightforward links to models of soil hydrology, plant growth and therefore OM inputs to soil. This kind of dynamic soil-plant model would encompass, for the first time, a complete description of all the physical feedback mechanisms determining organic C sequestration in soil.
We expanded our discussion of these important issues.

*The conclusions currently read more like a discussion. They could be sharpened to what readers should "take home" for their work from this study. What are the most important parameters and feedbacks that you think they need to consider in their experiments and studies?*

Response: Yes, we modified this section. In fact, in order to meet some of the other comments and suggestions from Dr. Wutzler and referee #2, we can see the need to include a short discussion section in the paper

*There are already models that let SOM decomposition affect soil structure. For example in the model of Ahrens et al. 2015 (see also Yu 2020 eq. S28a) SOM dynamics affects bulk soil density and soil volume and this in turn affects modeled concentrations, changes in soil volume, and transport processes. They applied the same concept of Federer 1993 as in the current manuscript, but incorporated many more processes so that this feedback was not explicitly discussed. The present manuscript additionally partitions micro-and mesoporosity and models protection by aggregation. A little comparison in the discussion or introduction would be nice.*

Response: We did include a comparison of our model with several previous models in the Introduction, but we had missed that Ahrens et al. and Yu et al. also model the effects SOM on bulk density (as Dr. Wutzler writes, this aspect of their model was not prominently discussed in the cited papers). We included a reference to Ahrens et al. and Yu et al..
Dr. Wutzler notes that in addition to the physical protection of SOM afforded by soil structure, we also model the effects of SOM on pore size distribution and water retention. As mentioned earlier, we consider that this is an important advance, because it enables subsequent links to models of soil water flow and plant growth. We emphasized this aspect of the model in more detail.

*P4L103: The authors argue that macropores probably are only a minor balance of SOM balance. Contrary, some researchers think, that macropores are a hot spot of SOM turnover and together with the rhizosphere are the most important places to study. Especially for systems with active earth worms this has been shown (e.g. Don et al. 2008).*

Response: It would be possible to extend the model to deal with C inputs to the macropore region, for example by root in-growth or the exploitation of surface litter by earthworms, although this would increase model complexity and introduce new parameters. We agree that this is something that should be explored in the future. We added some text on this in the final section (Discussion and Conclusions).

*eq. 7 and 11 seem to both add volume and additional pore space with addition of OM. In an alternative mind model putting dissolved organic matter or root exudates into soil would partly fill up existing pores. Please, add some explanation of assumptions to this part.*

Response: Yes, this is an interesting question, which goes to the heart of the model concept. Adding a mass of OM must increase the volume of OM, but it could either increase or decrease the pore volume and thus the total volume of the soil. The parameter that determines this is $f_{agg}$ (what we call the aggregation factor). If the addition of OM resulted in a net decrease of the pore volume, then $f_{agg}$ would take a negative value (the minimum value $f_{agg}$ could take is -1, if the added OM completely filled existing pore space, as Dr. Wutzler suggests it could). However, we can see from the data (see Figure 4) that $f_{agg}$ ~ 2, in other words, a volume of OM creates twice its own volume of pores. There are several mechanisms and processes (both biological and physical) underlying the generation of aggregation pore space, which would be difficult to model separately, so our model makes no attempt to do so. The only assumption underlying the model is the linear relationship between aggregation pore space and OM, something which is strongly supported by experimental evidence (see text at lines 104 – 106). We added some more explanatory text on this.

*I miss a paragraph how the model was integrated in time. I assume an explicit time (Euler forward) step much lower than the 5 years of distance between observations.*

Response: Yes, it was an explicit numerical solution with Euler integration and an annual time step. We added this missing information to the text.

*How did you track the changes in soil depth (eq. 12) in the comparison to data?*

Response: This was already explained in the paper. We simulated five soil layers, with variable thickness according to equation 12 (lines 351/352) and the difference in the total profile depth between the two treatments was compared with the difference in the soil surface elevation measured in 2009 (lines 336/337).

*Minor comments.*

*The discussion at p3L55 argues about soil structure affecting SOM dynamics. If one could show that it is not only affecting fast pools, then this argument could be made even stronger to affecting SOM stocks and soil carbon sequestration.*

Response: Yes, true. We feel that the experiments discussed at lines 64-66 (and other similar experiments) give very strong evidence for the protective effect of soil structure on slow OM pools.

*The font sizes in the figures are often very small, which makes it difficult to read the print-version.*

Response: Thanks for the hint. We increased the font size of the figures.

*eq.5 and 6: Why is there a factor of 1/2?*

Response: It follows from the definition of $k_{mix}$ as the intensity of mixing of the stored OM in the two pore classes at an annual time scale. It gives perfect mixing for $k_{mix} = 1$ year$^{-1}$.

*Please, check consistency of mathematical symbols. E.g. delta.z_min is sometimes written with min as subscript and sometimes with parenthesis (Table 1) denoting density gamma_o and gamma_m or gamma_org and gamma_min. F_text_mic or F_mic_text (fig. 6).*

Response: Thanks for pointing this out. We corrected these inconsistencies.

*p6L165: Parameter f_agg is introduced here. To my reading its quite an important parameter. I recommend explaining it (here or somewhere) in more detail. Does it correspond to the porosity of the volume occupied by organic matter?*

Response: Not exactly, although $f_{agg}$ is related to the porosity of organic matter, see equation 24 (please note that there has a typo in equation 24: $\phi_{mac}$ should replace $\phi_{min}$. We fixed this in the revised paper). $f_{agg}$ is simply the slope of the linear relationship that is assumed between the volumes of aggregation pore space and OM. We added some more explanation to the paper at line 174. We discussed the correspondence of the parameters of the Federer et al. (1993) model with our more fundamental derivation of essentially the same model at lines 217 – 234.

*eq 21-24: please, use a different symbols at the left hand side than in (19) and (20) to denote the quantities to use assumption of f_som = 0 r f_som = 1.*

Response: Thanks for the hint – we adapted the symbols.

*Sect 3.2: Given the 5 years interval of SOM measurements the non-identifyability of the fast turnover pool is expected. Could you think of additional observations or sub-experiments that could inform the shorter time scale?*

Response: Incubation experiments would be needed to quantify the dynamics of the young pool at shorter time scales. However, these kinds of experiments are usually conducted under controlled conditions in terms of water content and temperature, which makes it difficult to transfer the results to the field. Another approach would be to study the degradation of organic matter using litterbags (containing, for example, above-ground harvest residues). However, in the treatments that do not receive organic material, it can be assumed that the young pool consists of roots and rhizodeposition. Since it is difficult to quantify the amount of C that enters the soil via roots, it is also difficult to quantify its

degradation. We are not aware that these kinds of experiments have been carried out for the Ultuna frame trial.

*Sect 3.2. The mixing ratio was quite influential in Table 1. I assume in the identifiability analysis it correlated strongly with other parameters -which ones?*

Response: Yes, $k_{mix}$ correlated strongly with $k_y$, $k_o$, $F_{prot}$ and $F_{text(mic)}$. We added a table with this information (see the new Table 3).

*Could this lead to potential model simplifications?*

Response: In some model applications, possibly yes, but not if the users are interested in the influence of tillage or earthworm bioturbation on C sequestration

*P11L335: "root litter input was distributed uniformly across depth". What do you expect to be the effect of distribution root litter input with an exponentially decreasing profile? How do you treat partitioning of given total root input to the modeled top soil and the non-modeled lower depth?*

Response: In the ploughed (and sampled) horizon relevant to this study, there would be no effect at all, because of tillage mixing. Input to the topsoil was distributed uniformly because we assume efficient mixing by tillage. The non-modelled lower depth is not relevant to this paper.

*Fig 1: The dotted regions were not visible in my printout. Please adapt the pattern.*

Response: We replaced the dotted pattern by blank regions.

*Fig 2: The placing of the braces confused me. Vor micropores its at the maximum pore diameter for mesopores the lower boundary of the upper brace coincides with the blue line. To my understanding it should instead coincide with the red line at the upper diameter.*

Response: We checked the figure and it is OK. We don't really understand this comment

*Fig 3: Cannot read the subscripts in this figure. Please, adjust the font sizes. (Also in the other figures)*

Response: We adjusted the font sizes of Figure 3 and the other figures as well.

*Fig 8: I had to search for the difference between left and right panel. Please describe in the legend or make the font of the years 1997 or 2019 more prominent.*

Response: We added a title to both panels stating the respective year. We also added the information to the figure caption.

*Fig 9: Figure headings (bare fallow, manure) in addition to the legend would help the reader.*

Response: Thanks for the comment – we added headings to Figure 10 (former Figure 9) and also adjust the font size of the legend.

**Responses to referee #2**

*In this work, the authors propose a new framework to model soil organic matter turnover, which includes a two-way coupling between SOM storage and soil porosity. The model considers four pools of organic matter, with the dynamics described by four coupled differential equations. The novelty consists in using additional pools to divide the organic matter between micropore and mesopore soil regions, each one characterized by its own fluxes and decomposition rates. In my opinion such a model indeed can bring new insights about the dynamical feedback between soil physical properties and SOM decomposition, and can be an important contribution to the field. Although I find the paper interesting, I have some concerns. In particular I would have appreciated a more detailed discussion of the advantages of this new model. My recommendation is publication of this manuscript subject to a revision based on comments listed below.*

Response: We appreciate the positive feedback of reviewer#2. We included some more text on the advantages of this model compared to models that do not include interactions with soil physical properties in a final discussion and conclusions section

*1 - I find that the paper is in general well written, but the section with the description of the model is very confusing and needs to be improved. I would suggest to first write the full model including the feedback on porosity, and only afterward to follow with all the necessary derivations.*

Response: Thanks for the suggestion, which we have considered carefully. However, after trying out different options, we decided we would like to keep the current structure of the model description and derivation, which we are convinced will be easier to follow and understand for the reader. We think the model derivation is already clearly presented. However, meeting some of the changes requested by referee #1 and referee #3 should lead to further improvements in clarity.

*Also, it is not clear by looking at the equations which parameters are kept constant, one has always to search in the text.*

Response: Tables 1, 2 and 4 give information on the parameters that were kept constant and those that were varied in the sensitivity analysis and in the calibrations.

*One solution is to use upper case for functions and lower case for constants.*

Response: We are not sure we understand this suggestion, but the difference between parameters and functions is apparent from the equations themselves.

*Please also double check the notation, for example the density of mineral matter is ngamma_m on pg.6 and ngamma_min in all tables.*

Response: Thanks for the comment. We corrected this inconsistency (and also similar ones for the density of organic matter and the minimum layer thickness). We have double-checked the notation and cannot find any other errors.

*2 - The abstract states that the model successfully reproduces the soil water retention curves. I find this statement too strong due to the discrepancy of the curves for the year 1997.*

Response: Yes, we modified this sentence to make it clear that the model only successfully matched the water retention measurements obtained at the end of the experimental period in 2019.

*3 - I would like to see an extended discussion on the k_mix and F_prot, since these parameters are at the core of the discussed feedback. For large values of k_mix and F_prot nsim 1 the soil structure properties have to become less important to the dynamics of SOM turnover. Could the authors comment on this transition to the regime where the soil porosity becomes less relevant for the model outcome?*

Response: Yes, this is correct. We included a brief discussion of this in the revised paper

*I would also appreciate a short comment on the choice of the sampled range for the sensitivity analysis (and also the choices for calibration).*

Response: For the sensitivity analysis, we justified the choice of sampled ranges at lines $244 - 249$: The sampled ranges for calibration were based on previous experience with SOM models and some trial and error. The defined ranges for calibration could in theory influence the outcome of the calibration procedure if there are local minima in the goal function. However, we found that increasing the ranges shown in Table 3 had no impact on the results.

**Responses to referee #3**

*Meurer and colleagues describe a modified version of the ICBM model which is intended to describe a feedback between SOC formation and decomposition and its effects on bulk density and pore size distribution. While the premise of the study is very interesting it falls short in proving that the feedback between micropore space and SOC decomposition is needed to describe SOC dynamics.*

Response: We would like to thank the referee for his valuable feedback. The paper improved as a result.

*I would ask the authors to clarify and work on the following points:*

*- Please do a more thorough literature research: Before Federer et al. (1993) a couple authors have used equations similar to the Federer one, maybe even your Equation 20 (Adams, 1973; Rawls, 1983). These are just two examples - probably you can work your way backwards from here. Tranter et al. (2007) provide a good overview of the literature and show how soil texture affect mineral soil bulk density.*

Response: Thank you. Yes, Federer et al. were not the first to apply the model. We added this relevant literature in the text where currently we only mention Federer et al..

*- It would be interesting for the reader to see how much of bulk density changes is due to the difference in density between minerals and soil organic matter (mass effect), and how much due to changes in porosity (difference in porosity between minerals and soil organic matter?).*

Response: Yes, this is an interesting question. We interpret the second part of this sentence ("changes in porosity") to mean the effects of aggregation. This question can be answered by analyzing equation 20. We added a figure to the paper based on this equation showing how the relationship between bulk density and organic matter concentration varies with different values of $f_{agg}$ (the aggregation factor). The curve for $f_{agg}=0$ (i.e. no aggregation) shows that the different densities of organic and mineral matter have only a minor effect. Aggregation dominates the effects of organic matter on bulk density. We can mention here that that the simpler version of this bulk density model previously published does not allow for this kind of analysis, since it does not distinguish between these two effects.

*- Figure 7 suggests that the microporosity effect is minimal and the increase/decrease in bulk density is solely driven by the decrease/increase in SOC.*

Response: In this model application, all the parameters in equation 20 (for calculating bulk density) are considered as constants except for the organic matter content, so yes, the increase/decrease in bulk density is indeed solely driven by the decrease/increase in SOM.

*- Please provide some numbers how important SOC changes are for changes in microporosity.*

Response: Equation 25 shows how the time-varying SOC content and the (constant) soil textural pore space affect microporosity. The results of the sensitivity analysis suggest that the balance between microporosity and mesoporosity is most strongly determined by soil texture, which certainly agrees with past empirical experience. This was already discussed at lines 252 – 255.

*- You set F_prot a priori based on literature values. I think you have to provide more background to the reader how they were derived. SOC is then decomposing at a speed of 10 percent in micropores. Is this well constrained by experiments?*

Response: This value is based on a study published by Kravchenko et al. (2015) in which they used X-ray tomography to show that the decomposition rates of intra-aggregate particulate SOM were 3 – 15

times faster in the presence of connected networks of aerated soil pores > 13 μm in diameter than in the absence of such pores (see Introduction ll. 55 – 57). We chose the value for $F_{prot}$ based on this range. This was mentioned in the text at lines 349 – 351 and also in Table 3. Clearly, more experiments of this kind will help to better constrain this parameter value in the future. We discussed this at lines 441 – 444.

*- You use the term 'warm-up'. Please correct to spin-up.*

Response: We changed "warm-up" to "spin-up".

*-Please provide a complete list with all symbols and abbreviations. The reader can get lost in the amount of equations otherwise.*

Response: We added a list with symbols to the supplementary material.

---

## Author Response (AR2)

General comments from the authors:

Dear Dr. Richter

We would like to thank Dr. Wutzler for taking the time to read the paper again and provide more helpful feedback. The suggestions have helped us to further improve the paper.

We have responded to each of the suggestions below.

Comments by Thomas Wutzler:

The suggested comparison to a model variant without the feedbacks has been accomplished. Although the dynamic SOM-soil structure feedback does not much improve the predictive capacity at the Ultuna data, the model and the paper are worthwhile to publish.

Fig 2 now avoids the confusions.

Response: Fine! Thank you for making us aware of the confusion.

Thanks for the extended explanation of the important f_agg parameter to describe the porosity of organic matter. I was confused by the term "aggregate" suggesting to the formation of soil aggregates from both mineral and organic matter. Therefore I suggest to make this more clear also at other places and replace e.g. P8L233 phrases like "no aggregation" to "no additional pore space with organic matter").

Response: We are pleased that our additional explanation of the parameter $f_{agg}$ has helped. We have made the specific change suggested: we replaced "no aggregation" by "no additional pore space is generated due to the presence of organic matter". Please note, however, that we cannot completely avoid the use of the term "aggregation" in the paper, because this is what the model (implicitly) simulates (the parameter $f_{agg}$ is termed the *aggregation* factor). To address the concern of the referee on this point, we have improved the explanatory text in section 2.1 where we introduce the main concepts underlying the model, in order to clarify what we mean by "aggregation". We now write:

*"....in our model approach, which is based on the dynamics of soil pore space, "aggregation" is simply defined as the additional pore space in soil associated with the presence of organic matter. Based on empirical knowledge, we assume a linear relationship between this aggregation pore volume, $V_{agg}$, and the volume of soil organic matter (e.g. Emerson and McGarry, 2003; Boivin et al., 2009; Johannes et al., 2017). Thus, individual soil "aggregates" are not considered as explicit entities in this model."*

In Fig 5 there probably is a problem with the unit. (to 10% instead of only to 0.1% OM content?)

Response: Yes, thank you, we have corrected this.

P15L503: Please add "potentially": "it potentially enables … without re-calibrating". This still has to be shown. One may remind the reader that phi_min and f_agg can be computed from particle size distributions and bulk densities of organic matter and minerals.

Response: Yes, thank you, this is a very good point. We have done so.

P15L499 The nonlinear response of steady-state to organic matter inputs in itself is interesting to me. I would be happy about reading about the shape and a little discussion and implications of predictions compared to other soil models.

Response: Yes, we have added some more discussion, referring to a couple of other model approaches (based on concepts of C saturation) that also show non-linear responses of steady-state soil C to OM inputs.

[revised manuscript text omitted]
,  in a similar way to earlier models based on concepts of carbon saturation (e.g. Hassink and Whitmore, 1997; Stewart et al., 2007), the extended model described and tested here, which explicitly  incorporates two-way soil structure-$S$OM interactions, does not show such a linear response. This non-linearity of response of steady-state OM contents to OM inputs becomes stronger as the mixing between the pore regions becomes weaker.

Even though it may be possible to satisfactorily calibrate a simple OM model such as ICBM to time-series of OM measurements at one particular site, a model that explicitly incorporates soil structure-OM feedbacks has  some important advantages. For example, it potentially enables direct ("forward") simulations of the effects of soil structure and physical protection on OM turnover in contrasting soil types (e.g. sand vs. clay),  without having to resort to re-calibrating model parameters describing OM turnover for each soil, as was done, for example, by Poeplau et al. (2015). In our model, some of the key paramerters controlling physical protection can, in principle, be determined "*a priori*" from measurements. Thus, $\phi_{min}$ and $f_{agg}$ can be derived from paired data on soil organic matter contents and bulk density (equation 19), while $F_{text(mic)}$ can be calculated from particle size distributions (e.g. Arya and Heitman, 2015). In principle, our model also has a  
[revised manuscript text omitted]